# The interrelation of sleep and mental and physical health is anchored in grey-matter neuroanatomy and under genetic control

Masoud Tahmasian [1], Fateme Samea[1], Habibolah Khazaie[2], Mojtaba Zarei[1],
Shahrzad Kharabian Masouleh [3,4], Felix Hoffstaedter[3,4], Julia Camilleri[3,4], Peter Kochunov[5],
B. T. Thomas Yeo[6,7,8,9], Simon Bodo Eickhoff[3,4] & Sofie Louise Valk[3,4]✉

Humans need about seven to nine hours of sleep per night. Sleep habits are heritable, associated with brain function and structure, and intrinsically related to well-being, mental, and physical health. However, the biological basis of the interplay of sleep and health is incompletely understood. Here we show, by combining neuroimaging and behavioral genetic approaches in two independent large-scale datasets (HCP ($n = 1106$), age range: 22–37, eNKI ($n = 783$), age range: 12–85), that sleep, mental, and physical health have a shared neurobiological basis in grey matter anatomy; and that these relationships are driven by shared genetic factors. Though local associations between sleep and cortical thickness were inconsistent across samples, we identified two robust latent components, highlighting the multivariate interdigitation of sleep, intelligence, BMI, depression, and macroscale cortical structure. Our observations provide a system-level perspective on the interrelation of sleep, mental, and physical conditions, anchored in grey-matter neuroanatomy.

[1] Institute of Medical Science and Technology, Shahid Beheshti University, Tehran, Iran. [2] Sleep Disorders Research Center, Kermanshah University of Medical Sciences, Kermanshah, Iran. [3] Institute of Neuroscience and Medicine (INM-7: Brain and Behaviour), Research Centre Jülich, 52425 Jülich, Germany. [4] Institute of Systems Neuroscience, Heinrich Heine University Düsseldorf, 40225 Düsseldorf, Germany. [5] Maryland Psychiatric Research Center, University of Maryland School of Medicine, Baltimore, MD 21201, USA. [6] Department of Electrical and Computer Engineering, Clinical Imaging Research Centre, N.1 Institute for Health and Memory Networks Program, National University of Singapore, Singapore 119077, Singapore. [7] Athinoula A. Martinos Center for Biomedical Imaging, Massachusetts General Hospital, Charlestown, MA 02114, USA. [8] Centre for Sleep and Cognition, National University of Singapore, Singapore 119077, Singapore. [9] NUS Graduate School for Integrative Sciences and Engineering, National University of Singapore, Singapore 119077, Singapore. ✉email: svalk@fz-juelich.de

Sleep plays an active role in providing adaptive physiological functions[1], consolidating and retaining new memories[2], metabolite clearance[3], hormones' secretion[4], and synaptic hemostasis[5]. The National Sleep Foundation suggests 7–9 h of sleep per night for adults (18–64) and 7–8 h for older adults (65+). For school aged children (6–13 years) this is 9–11 h, and for teenagers 8–10 h[6]. However, people in modern societies are suffering from inadequate sleep and its consequences[6]. Sleep loss is associated with impairment in cognitive performance, motor vehicle accidents and poor quality of life[7,8]; and contributes to heightened socioeconomic burden[9,10]. Beyond the quantity of sleep (sleep duration), quality of sleep includes sleep onset latency (i.e., time between going to bed and falling asleep), sleep efficiency (i.e., the percentage of time in bed during which someone is asleep), sleep disturbances, use of sleeping medication, and daytime dysfunction, all interacting with individual health and well-being[11,12]. Of note, it has been revealed that poor sleep quality is associated with higher rate of depressive symptoms in healthy subjects[13,14]; and sleep disturbances are common in mood (e.g., major depression) and cognitive disorders[15,16].

Individual differences in sleep behaviors are heritable[17–19]; and various genetic, metabolic, behavioral, and psychological risk factors have been suggested for the development and maintenance of poor sleep quality and sleep disorders[20–22]. For example, genome wide association studies have associated insomnia disorder to structure of the striatum, hypothalamus, and claustrum, where gene expression profiles show association with the genetic risk profile of such individuals[23,24]. Moreover, sleep can have a bidirectional relation with health. Not only is sleep disturbance linked with hypertension, diabetes, and obesity[25,26], but also depressive symptoms, physical illness, and fatigue were reported as associated factors for both poor sleep quality and short sleep duration[27,28]. A meta-analysis reported that insomnia disorder is associated with alterations in widespread brain structure and function[29]. In addition, other neuroimaging meta-analyses have implicated structural and functional abnormalities in the hippocampus, amygdala, and insula in patients with sleep apnea[30] and have indicated convergent functional brain alterations in the inferior parietal cortex and superior parietal lobule, following acute sleep deprivation[31]. Moreover, white matter integrity underlying prefrontal areas has been associated with sleep duration and sleep quality[32–34]. Lastly, lower prefrontal gray matter volume has been associated with greater sleep fragmentation in older individuals[35].

Importantly, it has been demonstrated that macroscale gray matter neuroanatomy is heritable[36–38], indicating part of the variance in brain structure can be related to additive genetic effects. Indeed, genetic factors influence cortical thickness in a systematic fashion where both functional and geometric constraints influence genetic correlation between and within brain systems[39,40]. Recent studies have indicated that phenotypic correlation between cortical thickness and intelligence, as well as BMI, is driven by additive genetic factors[41–43] suggesting a shared genetic basis of cortical thickness and non-brain traits. This raises the question whether the interrelation of sleep, mental, and physical health can be linked to the shared neurobiological mechanisms; and whether these relationships are driven by shared genetic factors.

To answer this question, we combined structural neuroimaging data from two independent samples: the Human Connectome Project (HCP unrelated sample $n = 424$) and the enhanced NKI Rockland sample (eNKI: $n = 783$) to explore whether the interrelation of sleep, mental, and physical health can be linked to a shared macroscale neurobiological signature. The HCP sample consists of young adults only, whereas the eNKI sample consists of adolescents, younger and older adults, enabling us to evaluate the generalizability of the interrelation of sleep, health and local brain structure. We conducted genetic correlation analysis in the

complete HCP sample ($n = 1105$), which included twins and siblings. Sleep variation was assessed using the Pittsburg Sleep Quality Index (PSQI)[11], a widely used questionnaire summarizing self-reported indices of sleep. Our main measures of interest were sleep quantity (self-reported sleep duration) and global sleep quality (total PSQI score), as previous work has associated these factors with brain structure[44,45] and genetic variation[46]. Based on previous literature[7,8,25–28] and data-driven phenotypic correlations in the HCP sample, we selected BMI, intelligence and depression scores to evaluate the potential existence of a shared neuroanatomical basis of sleep and mental and physical aptitudes. In the HCP sample, intelligence was summarized as Total Cognitive Score, based on the NIH Toolbox Cognition[47], whereas in the eNKI sample, intelligence was measured using the Wechsler Abbreviated Scale of Intelligence (WASI-II)[48]. Depression was measured using the ASR depression DSM-oriented scale for Ages 18–59[49] in the HCP sample. In the eNKI sample the Beck Depression Inventory (BDI–II) was used. BMI was calculated at weight/squared (height) in both datasets. Based on previous knowledge, we expected to observe phenotypic relationships between sleep duration/quality and markers of mental and physical health. Moreover, we expected to observe a phenotypic relation between sleep and local gray matter structure.

Our analyses revealed a phenotypic relationship between sleep and depression, BMI, and intelligence in both the HCP and the eNKI sample. Subsequently, we demonstrated our sleep markers, depression, BMI, and intelligence were heritable and we observed a genetic correlation between sleep quantity and quality, BMI, and intelligence in the twin-based HCP sample, indicating that sleep hygiene displays pleiotropy with these factors in the mentioned sample. Analysis of heritability and genetic correlation were performed with maximum likelihood variance-decomposition methods using Sequential Oligogenic Linkage Analysis Routines (www.solar-eclipse-genetics.org; Solar Eclipse 8.4.0.). Heritability ($h^2$) is the total additive genetic variance and genetic ($\rho_g$) correlations were estimated using bivariate polygenic analysis. Using an atlas-based approach to summarize cortical thickness[50], we observed local associations between sleep duration and cortical structure in both samples which were, in part, driven by additive genetic factors. Post-hoc analysis indicated that variance in intelligence and BMI also related to thickness in areas associated with sleep duration. Subsequently, based on our observation that sleep relates to BMI, intelligence, and depression, as well as to cortical thickness, we performed partial least squares (PLS) analysis, in order to identify latent relationships between these factors. PLS is a multivariate data-driven approach, enabling simultaneous linking of behavioral measures to brain structure. We identified two robust latent factors, spanning distinct neurocognitive dimensions. Using the twin-structure of HCP, we observed these factors were heritable and their relation driven by shared genetic effects. Taken together, the current study highlights the interrelation of sleep, mental and physical health, which is reflected by shared neurobiological signatures.

## Results

**Data samples.** We studied two independent samples from openly-shared neuroimaging repositories: HCP and eNKI. HCP (http://www.humanconnectome.org/) comprised data from 1105 individuals (599 females), 285 MZ twins, 170 DZ twins, and 650 singletons, with mean age 28.8 years (SD = 3.7, range = 22–37). For phenotypic analysis, we selected unrelated individuals, resulting in a sample of 424 (228 females) individuals with a mean age of 28.6 years (SD = 3.7, range = 22–36). Our second sample was based on the eNKI sample, made available by the Nathan-Kline Institute (NKY, NY, USA)[51]. This sample consisted

**Table 1 Phenotypic and genetic correlations between sleep and depression, BMI, and IQ.**

**Sleep duration (h2 = 0.24 ± 0.06)**

| Sample | Depression (h2 = 0.24 ± 0.06) | BMI (h2 = 0.68 ± 0.04) | IQ (h2 = 0.66 ± 0.04) |
|---|---|---|---|
| HCP (unrelated sample) | ($n = 419$) −0.09 [−0.19 0.00], $p = 0.06$ | ($n = 424$) −0.11 [−0.21 −0.02], $p < 0.025$* | ($n = 418$) 0.11 [0.01 0.19], $p < 0.05$* |
| eNKI | ($n = 782$) −0.16 [−0.24 −0.09], $p < 0.001$** | ($n = 757$) −0.17 [−0.24 −0.09], $p < 0.001$** | ($n = 783$) 0.11 [0.04 0.18], $p < 0.005$* |
| HCP (total sample) | ($n = 1105$) −0.07 [−0.13 −0.02], $p < 0.025$* | ($n = 1112$) −0.14 [−0.19 −0.08], $p < 0.0001$** | ($n = 1096$) 0.09 [0.03 0.15], $p < 0.005$* |
| Genetic correlation (HCP) | 0.17(0.20), $p > 0.1$ | −0.33 (0.11), $p < 0.005$* | 0.42 (0.11), $p < 0.0001$** |
| Environmental correlation (HCP) | −0.16(0.06), $p < 0.01$* | 0.01 (0.07), $p > 0.1$ | 0.19 (0.06), $p < 0.003$** |

**Global sleep quality (h2 = 0.12 ± 0.06)**

| Sample | Depression | BMI | IQ |
|---|---|---|---|
| HCP (unrelated sample) | ($n = 419$) 0.37 [0.29 0.45], $p < 0.0001$** | ($n = 424$) 0.14 [0.04 0.23], $p < 0.005$* | ($n = 418$) −0.07 [−0.16 0.03], $p > 0.1$ |
| eNKI | ($n = 782$) 0.31 [0.25 0.38], $p < 0.0001$ ** | ($n = 757$) 0.09 [0.02 0.17], $p < 0.01$* | ($n = 419$) −0.09 [−0.16 −0.02], $p < 0.01$* |
| HCP (total sample) | ($n = 1112$) 0.35 [0.30 0.40], $p < 0.0001$** | ($n = 1112$) 0.10 [0.04 0.16], $p < 0.001$** | ($n = 1096$) −0.10 [−0.16 −0.04], $p < 0.002$* |
| Genetic correlation (HCP) | 0.32(0.26), $p > 0.1$ | 0.41 (0.16), $p < 0.025$* | −0.59 (0.20), $p < 0.0001$** |
| Environmental correlation (HCP) | 0.38(0.05), $p < 0.0001$** | 0.03 (0.07), $p > 0.1$ | 0.17 (0.06), $p < 0.007$** |

We performed phenotypic (HCP unrelated sample, eNKI sample, HCP total sample) and genetic correlation (HCP total sample) analysis of the association between sleep duration and global sleep quality on the one hand, and depression, BMI, and IQ on the other, including 95% confidence intervals. Asterisks (**) indicates FDR $q < 0.05$ and asterisk (*) indicates association at trend−level $p < 0.05$. Sample sizes are reported for each analysis.

of 783 (487 females) individuals with mean age of 41.2 years (SD = 20.3, range = 12–85), enabling us to identify life-span relations between sleep, brain structure and behavior. Details on the sample characteristics can be found in the Methods section.

**Relation between sleep, mental and physical health**. First, we sought to evaluate whether our measures of mental and physical status are related to sleep quantity and quality. Here, we correlated sleep duration and global sleep quality to phenotypic variation in cognition, mental, and physical health (for selection of markers see Supplementary Table 1). This data-driven analysis in the HCP phenotypic data revealed that cognitive, mental and physical phenotypic variation have a strong relation to variation in sleep (Supplementary Table 2 and Supplementary Table 3). Given the marked role of both depression and BMI on both sleep duration and global sleep quality, we selected these as phenotypes of interest for further analyses. As several cognitive factors were related to sleep duration and sleep quality, we selected general intelligence in this study, as this marker has been shown to be highly heritable and consistently relates to brain structure[42]. Next, we demonstrated that depression, IQ, and BMI have moderate phenotypic inter-correlations in unrelated HCP, eNKI, as well as full HCP samples (Supplementary Table 4). Evaluating the relation between sleep and our selected markers in eNKI, in addition to HCP, we observed that, sleep duration had a consistent negative phenotypic relation to both BMI and depression, and a positive relation to IQ (Table 1).

Taking advantage of the pedigree-structure of the full HCP sample, we observed that depression, IQ, and BMI were all heritable (Table 1); and we observed a negative genetic correlation between BMI and IQ ($\rho_g = -0.27$, $p < 0.0001$) (Supplementary Table 4). Moreover, sleep duration was heritable (h2 = 0.24, $p < 0.001$), and phenotypic correlations were mirrored by genetic correlations. We observed sleep duration to show a positive genetic correlation with IQ ($\rho_g = 0.42$, $p < 0.0001$), but negative with BMI ($\rho_g = -0.33$, $p < 0.005$) (Table 1). Depression showed a weak environmental correlation with sleep duration ($\rho_e = -0.16$, $p < 0.01$).

Global sleep quality showed comparable relations to depression, IQ and BMI, with strong phenotypic correlation between poor sleep quality (higher total PSQI score) and higher depression and BMI scores, as well as between poor sleep quality and lower IQ across samples (Table 1). Global sleep quaity was also influenced by additive genetic effects (h2 = 0.12, $p < 0.05$), but less so than

sleep duration. Phenotypic correlations were paralleled by genetic correlations, were poor sleep quality were genetically correlated with lower IQ ($\rho_g = -0.59$, $p < 0.0001$) and higher BMI ($\rho_g = 0.41$, $p < 0.025$). Again, depression only showed environmental correlation with global sleep quality ($\rho_e = 0.38$, $p < 0.0001$) (Table 1).

**Phenotypic correlation between sleep and brain structure in two independent samples**. Next, we evaluated the phenotypic relation between sleep indices (global sleep quality and sleep duration) and cortical thickness in both the unrelated subsample from HCP ($n = 424$) and eNKI ($n = 783$). Behaviorally, we observed a strong negative correlation (Spearman $r = -0.51$ [−0.59 −0.44], $p < 0.0001$) between global sleep quality and sleep duration (Fig. 1a). Correlation of sleep indices with brain structure demonstrated a negative link between left superior frontal thickness (area 6d2 and pre-supplementary motor area) and sleep duration (Spearman $r = -0.1$, FDR $q < 0.02$, Fig. 1b), that remained significant when controlling for self-reported depressive symptoms, as well as intake of sleep medications, intelligence, and BMI. Global sleep quality did not relate to local variations in cortical thickness (Fig. 1c). When evaluating the relationship in the complete HCP sample, including twins and siblings, we observed only a trending relation between sleep duration and cortical thickness (Supplementary Fig. 1).

In eNKI, we replicated the negative behavioral correlation between sleep duration and global sleep quality (Spearman $r = -0.53$ [−0.58 −0.47], $p < 0.0001$) (Fig. 1d). Though we again found no relation between global sleep quality and cortical brain structure (Fig. 1f), sleep duration showed a positive link between bilateral inferior temporal regions (left: Spearman $r = 0.13$, FDR $q < 0.02$, right: Spearman $r = 0.12$, FDR $q < 0.02$) and right occipital cortex (Spearman $r = 0.14$, FDR $q < 0.02$) (Fig. 1e). Findings remained significant when controlling for self-reported depressive symptoms, as well as intake of sleep medications, intelligence, and BMI.

In both samples, most individuals (>65%) slept less than the recommended 7–9 h (Supplementary Table 5) and only a small proportion of both samples (<9%) slept 9 h or more. Post-hoc analysis evaluating the linear relationship between short and long sleep duration and local brain structure replicated overall effects between sleep duration and local brain structure in individuals who reported to sleep less than 9h per night (short-to-normal sleep duration), but not in individuals sleeping more than 7h per night (normal-to-long sleep duration) (Supplementary Fig. 2). As

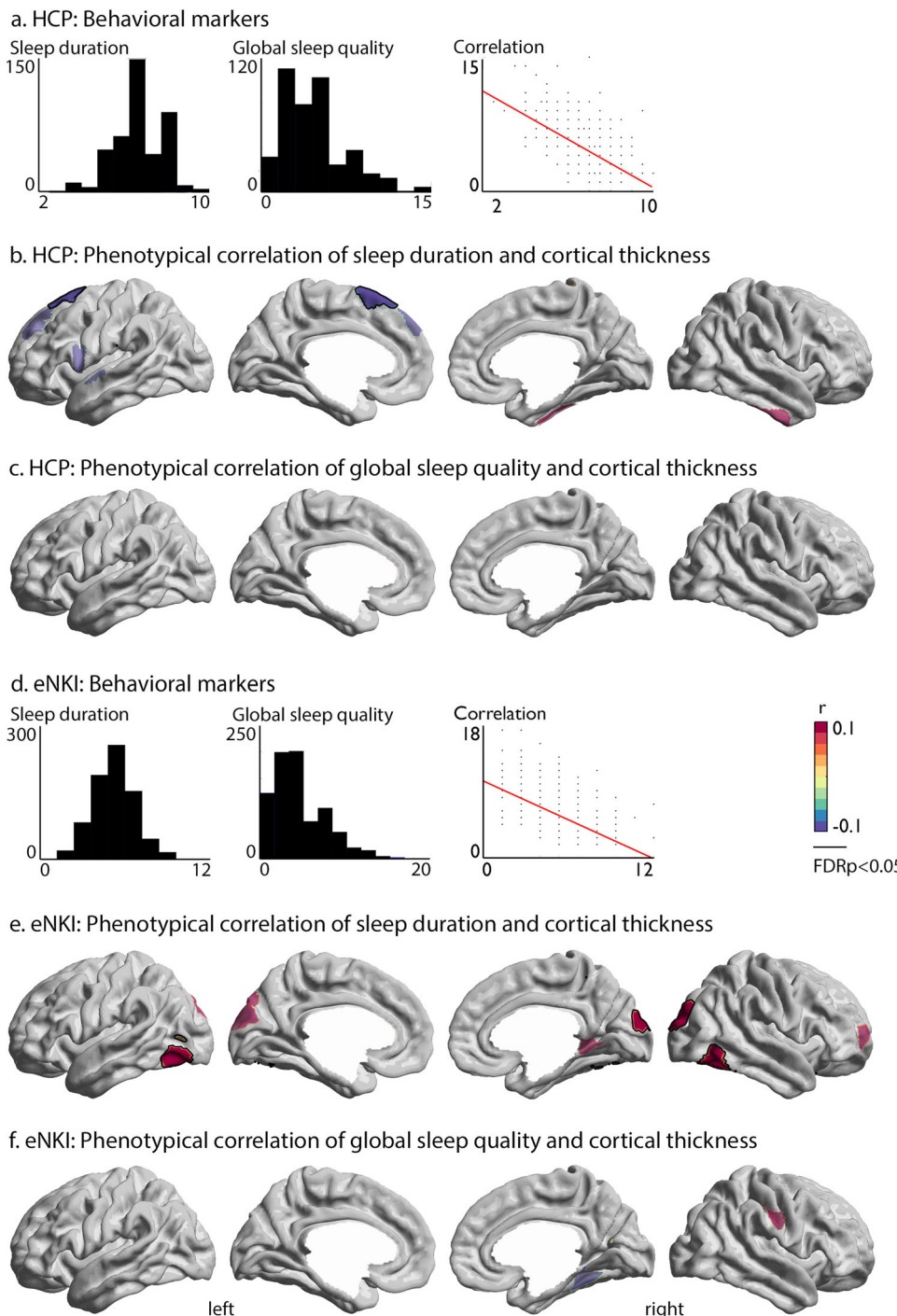

**Fig. 1 Patterns of phenotypic correlation between sleep duration and cortical thickness in HCP and eNKI samples. a** Distribution of variables in the unrelated HCP subsample; **b**, **c**. Phenotypic correlation of sleep duration/global sleep quality and cortical thickness; **d** Distribution of variables in the eNKI sample, as well as the correlation between sleep duration and global sleep quality score; **e**, **f** Phenotypic correlation of sleep duration/global sleep quality and cortical thickness. Red indicates a positive relationship, whereas blue indicates a negative phenotypical relationship between sleep and brain structure. Whole-brain findings were corrected for multiple comparisons using FDR correction ($q < 0.05$, black outline). Significant associations between sleep indices and brain structure have black outline, whereas trends ($p < 0.01$) were visualized at 60% transparency.

the eNKI sample had a broad age range from 12 to 85 years of age, we performed several stability analyses to evaluate the relationship between sleep duration and brain structure in youths, adults and elderly populations (Supplementary Table 6). Here, we did not observe differential sleep duration effects in each sub-group, as well as differences between age-groups (Supplementary Fig. 3).

**Replication analysis of correspondence between sleep duration and cortical thickness.** As we found divergent local phenotypic correlations between sleep duration and cortical thickness in two large-scale independent samples, we evaluated the inconsistencies across samples more precisely. Indeed, post-hoc analysis indicated that local effects of phenotypic correlations varied strongly

**Table 2 Inconsistency of associations between sleep duration and cortical thickness across samples and analyses.**

**Phenotypic correlation, Fig. 1b (HCP)**

| *Left superior frontal gyrus* | | *p* |
|---|---|---|
| HCP (unrelated sample) ($n = 424$) | $r = -0.19$ | 0.00007** |
| eNKI ($n = 783$) | $r = -0.03$ | 0.40 |
| HCP (total sample) ($n = 1113$) | $r = -0.10$ | 0.0013* |
| Genetic correlation (HCP) | $\rho_g = -0.27$ | 0.035* |
| Environmental correlation (HCP) | $\rho_e = 0.02$ | 0.97 |

**Phenotypic correlation, Fig. 1e (eNKI)**

| *Left inferior temporal cortex* | | *p* |
|---|---|---|
| HCP (unrelated sample) ($n = 424$) | $r = 0.09$ | 0.07 |
| eNKI ($n = 783$) | $r = 0.13$ | 0.0001** |
| HCP (total sample) ($n = 1113$) | $r = 0.04$ | 0.14 |
| Genetic correlation (HCP) | $\rho_g = -0.09$ | 0.67 |
| Environmental correlation (HCP) | $\rho_e = 0.06$ | 0.27 |
| *Right occipital cortex* | | *p* |
| HCP (unrelated sample) ($n = 424$) | $r = 0.07$ | 0.13 |
| eNKI ($n = 783$) | $r = 0.14$ | 0.0001** |
| HCP (total sample) ($n = 1113$) | $r = 0.04$ | 0.20 |
| Genetic correlation (HCP) | $\rho_g = 0.28$ | 0.03* |
| Environmental correlation (HCP) | $\rho_e = -0.08$ | 0.19 |
| *Right inferior temporal cortex* | | *p* |
| HCP (unrelated sample) ($n = 424$) | $r = 0.05$ | 0.30 |
| eNKI ($n = 783$) | $r = 0.12$ | 0.0006** |
| HCP (total sample) ($n = 1113$) | $r = 0.004$ | 0.90 |
| Genetic correlation (HCP) | $\rho_g = 0.38$ | 0.02* |
| Environmental correlation (HCP) | $\rho_e = -0.11$ | 0.06 |

Cross-sample replication of FDR-corrected ROIs from phenotypic correlational analysis in Fig 1. Asterisks (**) indicates to significant correlation ($q < 0.05$) and asterisk (*) indicates association at trend-level $p < 0.05$. Sample sizes are reported for each analysis.

**Table 3 Phenotypic associations between sleep indices and cortical thickness are mirrored by genetic correlations.**

**Phenotypic correlation, Fig. 1b (HCP)**

| *Left superior frontal gyrus* | | *p* |
|---|---|---|
| HCP (total sample) ($n = 1113$) | $r = -0.10$ | 0.0013* |
| Genetic correlation (HCP) | $\rho_g = -0.27$ | 0.035* |
| Environmental correlation (HCP) | $\rho_e = 0.02$ | 0.97 |

**Phenotypic correlation, Fig. 1e (eNKI)**

| *Left inferior temporal cortex* | | *p* |
|---|---|---|
| HCP (total sample) ($n = 1113$) | $r = 0.04$ | 0.14 |
| Genetic correlation (HCP) | $\rho_g = -0.09$ | 0.67 |
| Environmental correlation (HCP) | $\rho_e = 0.06$ | 0.27 |
| *Right occipital cortex* | | *p* |
| HCP (total sample) ($n = 1113$) | $r = 0.04$ | 0.20 |
| Genetic correlation (HCP) | $\rho_g = 0.28$ | 0.03* |
| Environmental correlation (HCP) | $\rho_e = -0.08$ | 0.19 |
| *Right inferior temporal cortex* | | *p* |
| HCP (total sample) ($n = 1113$) | $r = 0.004$ | 0.90 |
| Genetic correlation (HCP) | $\rho_g = 0.38$ | 0.02* |
| Environmental correlation (HCP) | $\rho_e = -0.11$ | 0.06 |

Genetic and environmental correlation between sleep and thickness in FDR-corrected ROIs from phenotypic correlational analysis in Fig. 1. Asterisks (**) indicates to significant correlation ($q < 0.05$) and asterisk (*) indicates association at trend-level $p < 0.05$.

in magnitude across samples in phenotypic analysis (Table 2). At the same time, we observed a high overlap between spatial distribution of phenotypic correlations between sleep duration, but not global sleep quality, and cortical thickness across samples and sub-samples, indicating that the direction of sleep thickness associations is similar across both samples (Supplementary Table 7). This suggests that the relation between sleep and cortical thickness might be robust at the inter-regional level rather than in local effects only. In addition, we observed that both intelligence and BMI related to local thickness associated with sleep duration (Supplementary Table 8), suggesting that sleep, intelligence and BMI are dependent on overlapping macro-anatomical structures.

**Phenotypic correlations between sleep and cortical thickness are driven by additive genetic effects**. Next, we explored whether phenotypic correlations between sleep duration and cortical thickness were mirrored by additive genetic effects using the twin-structure of the HCP dataset. First, we confirmed that cortical thickness was heritable in this sample (Supplementary Fig. 4, Supplementary Table 9). Second, we assessed whether phenotypic correlations observed in Fig. 1 were driven by shared additive genetic effects. We found that both frontal cortex (based on HCP), as well as right occipital cortex and right inferior temporal cortex (based on eNKI) showed a trend-level genetic correlation ($p < 0.05$) with sleep duration (Table 3). Using a whole-brain approach, we identified a negative genetic correlation between sleep duration and bilateral frontal cortices thickness, mainly in the bilateral superior frontal gyrus and frontal pole, areas p32 and Fp2 (left: $\rho_e = 0.12$, $p < 0.06$, $\rho_g = -0.46$, FDR $q < 0.025$; right: $\rho_e = 0.15$, $p < 0.01$, $\rho_g = -0.46$, FDR $q < 0.025$) (Supplementary

Fig. 1). Findings were robust when controlling for intelligence, BMI or depression score (Supplementary Table 10). Frontal regions showing genetic correlation with sleep in the HCP sample did not show an association with sleep in the eNKI sample (Supplementary Table 11). At the same time, we observed patterns of genetic correlation to reflect phenotypic correlation at the whole brain level within the HCP sample and sub-sample, and in the eNKI sample (Supplementary Table 7). Last, though we did not observe a genetic correlation between global sleep quality and brain structure, we identified an environmental relation between global sleep quality and left precentral thickness (Spearman $r = 0.01$, $\rho_e = 0.22$, $p < 0.0002$, $\rho_g = -0.64$, $p < 0.0003$) (Supplementary Fig. 5).

**Latent relation between sleep, brain and behavior**. As we observed (1) phenotypic and (2) genetic correlations between sleep, intelligence, BMI, and, in part, depression, as well as (3) an inconsistent relation between sleep duration and cortical thickness, we utilized a multivariate data-driven approach to evaluate the latent relationship between sleep, intelligence, BMI and depression on the one hand, and cortical thickness on the other (Fig. 2). Indeed, it has been suggested multiple comparison corrections in mass univariate analysis may result in missing effects and thus inconsistencies in the results and a more comprehensive picture of the associations could be gained by a multivariate approach[52]. Here, our primary analysis sample is the eNKI sample, as this enables us to replicate and evaluate phenotypic and genetic correlations between latent structures using the full HCP sample.

In the eNKI sample, we observed two latent relations between our behavioral phenotypes and cortical thickness, controlling for effects of age, sex and global thickness, explaining, respectively, 41% of the shared variance (first latent component; $p < 0.001$, association between behavior and brain saliencies: Spearman $r = 0.38$), and 25% of the shared variance (second latent component; $p < 0.01$, association between behavior and brain saliencies: Spearman $r = 0.29$). The first component had a positive relation with both sleep duration (Spearman $r = 0.49$) and intelligence (Spearman $r =$

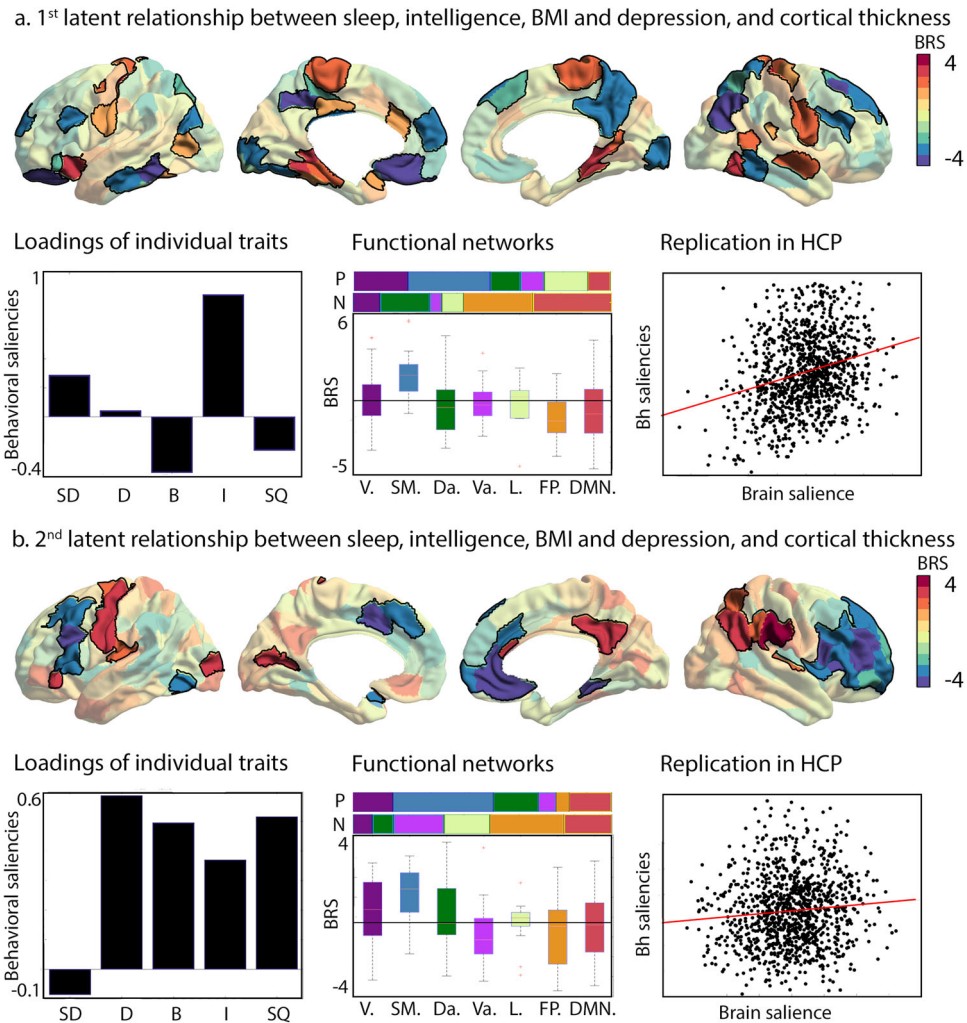

**Fig. 2 Two latent dimensions of cortical macrostructure and components of sleep, mental, and physical health. a** Bootstrap ratio of the first brain saliency that showed significant robustness, where parcel-wise saliencies of BSR > 2 are highlighted. Red indicates a positive association whereas blue indicates a negative association; Loadings of the individual traits (SD: Sleep duration, D: Depression, B: BMI, I: Intelligence, SQ: Sleep quality); Relative distribution of positive(P) and negative (N) -2>BSR > 2 scores per functional networks[102], and average BSR in functional networks[102] (V = visual, SM = sensorimotor, Da = dorsal-attention, Va = ventral attention, L = limbic, FP = frontopolar, DMN = default mode network); Replication of brain–behavior saliency association in the HCP sample; and **b** Relation between brain and behavioral saliencies in HCP sample of the second brain saliency; Loadings of the individual traits; Relation to functional networks[102] and; Relation between brain and behavioral saliencies of second factor in the HCP sample.

0.83), and a negative relation with sleep quality (Spearman $r = -0.43$), BMI (Spearman $r = -0.46$) and depression (Spearman $r = -0.21$). The brain saliency loadings were positively (bootstrap ratio > 2) associated sensory-motor areas, as well as superior temporal areas, and parahippocampal structures, and negatively with lateral and medial frontal cortex, as well as inferior temporal lobe, precuneus, and posterior parietal cortex. Further qualifying the brain saliency, we observed that positive relations were mainly in visual, sensorimotor and limbic functional networks, whereas negative relations were predominantly located in the dorsal-attention, fronto-parietal and default-mode networks. Replicating the association in HCP using the behavioral and brain loadings, we identified a relation between latent brain and behavioral factors in this sample as well (component 1: Spearman $r = 0.25$, $p < 0.001$). Moreover, both brain and behavioral latent factors were heritable (brain saliency: h2 ± std = 0.63 ± 0.04; behavior saliency: h2 ± std 0.76 ± 0.03) and showed genetic correlation ($\rho_e = -0.07 \pm 0.07$, $p = $ ns, $\rho_g = 0.38 \pm 0.05$, $p < 0.0001$). The first brain-behavior component seems to reflect a positive-negative axis of behavior, relating high sleep quantity to positive behaviors whereas low sleep quality related negatively to this factor (Fig. 2a).

The second component related positively to depression (Spearman $r = 0.70$), BMI (Spearman $r = 0.50$), intelligence (Spearman $r = 0.16$), and reduced sleep quality (Spearman $r = 0.69$), and negatively to sleep duration (Spearman $r = -0.43$). Positive brain loadings (bootstrap ratio>2) were located in the left sensorimotor areas, right precuneus, and right parietal areas. Negative loadings (bootstrap ratio < $-2$) were located in left dorsolateral areas, left mid-cingulate, right dorsolateral frontal cortex, and left anterior-mid cingulate. Qualitative analysis revealed positive loading were predominantly in sensorimotor, visual, dorsal attention and default networks, whereas negative loadings were associated with the fronto-parietal, ventral attention, limbic and default networks. Again, we replicated this association in HCP using the behavioral and brain loadings (Spearman $r = 0.10$, $p < 0.002$). Both brain and behavioral saliency of the second component were heritable (brain saliency: h2 ± std = 0.72 ± 0.03; behavior saliency: h2 ± std 0.51 ±

0.05) and showed genetic correlation ($\rho_e = -0.11 \pm 0.07$, $p = $ ns, $\rho_g = 0.22 \pm 0.07$, $p < 0.0001$). This time, sleep quality, depression, BMI, and intelligence showed positive latent relations, but duration had a negative relation to the behavioral saliency, suggesting that sleep quality has both positive and negative relationships to intelligence (Fig. 2b).

## Discussion

Sleep is key for normal human functioning and associated with brain structure and function. At the same time, individual differences in sleeping behavior are heritable and have substantial overlap with cognition, physical, and mental health. This raises the question whether shared variance in sleep, intelligence, BMI, and depression could be due to a shared relationship to macroscale gray-matter anatomy. Here, we combined computational approaches from behavioral genetics and big-data neuroimaging to evaluate the interrelation between sleep, macroscale brain structure, and mental and physical health. Indeed, in two large-scale independent samples, we observed that sleep duration, as well as global sleep quality, had a phenotypic relation with intelligence and BMI, which was mirrored by additive genetic effects. Depression showed only a phenotypic correlation with sleep. Following, we demonstrated that sleep duration, but not global sleep quality, had a (inconsistent) relation with local variance in cortical thickness in two samples. Three out of four phenotypic relations between sleep duration and local cortical thickness were driven by additive genetic factors. At the same time, both intelligence and BMI related to variance in cortical thickness in these regions, suggesting that these factors might have an overlapping neuroanatomical basis. Consistent with these results, a comprehensive multivariate analysis revealed two robust and heritable signatures, highlighting shared relationships between macroscale anatomy and sleep, intelligence, BMI and depression. Both components featured brain structures in both unimodal and heteromodal association areas, and underlined the embedding of nocturnal behavior in daytime functioning. Collectively, our multi-sample approach provides evidence that sleep is intrinsically interrelated with macroscale gray matter structure, mental, and physical health.

Our observations highlight the key relation between intelligence, mental and physical health and sleep profile in healthy subjects. Previous work has implicated the important role of sleep on life functioning, such as cognitive performance and quality of life[7,8], as well as higher rate of depressive symptoms[13,14], hypertension, diabetes, and obesity[25,26]. Indeed, clear associations of sleep, cognitive performance and behavioral problems have been observed in children[53], adults[54], and elderly[44]. It has been revealed that short-term sleep deprivation has a deleterious effect on a broad range of cognitive domains[54] and short sleep duration is associated with poor overall IQ /cognitive performance in healthy children[55].

There are various hypotheses on the biological processes underlying the important role of sleep in the neuronal processing of information and consequently mental processing. The trace reactivation or replay hypothesis[56,57] suggests that sleep helps memory consolidation through reactivation of traces of neuronal activity patterns, encoding information. The synaptic homeostasis hypothesis proposes that sleep is necessary to counterbalance the increase of synaptic connectivity[5]. Converging evidence suggests a role of sleep in maintaining functional integrity of the fronto-parietal networks, that support sustained attention[58,59], as well as default mode network[60], which is a brain network, implicated in task-unrelated thought. Indeed, in our multivariate analysis, we observed a shared relation of intelligence and sleep with cortical thickness in these networks. Importantly, we observed a positive

phenotypic relationship between amount of sleep and domain-general cognitive skill in both the HCP sample, consisting of young healthy adults, and in the eNKI sample, which included a broad age-range with children, adults and elderly. Of note, though the measurement of domain-general cognition was not consistent across two samples, as we used NIH Toolbox Cognition in HCP and the WASI-II in eNKI, both tests have been validated for different age-ranges[47,48]. Further studies are needed to uncover the causal and longitudinal relationship between sleep and cognitive skill across the life-span.

At the same time, our work highlights that inadequate sleep is linked with increased BMI. Previously, it has been shown that high BMI is associated with abnormal sleep duration and vice versa[61]. Short term sleep restriction is associated with impaired glucose metabolism, dysregulation of appetite, and increased blood pressure, and prospective studies found increased risk of weight gain associated with inadequate sleep[62,63]. In the same vein, various studies have related BMI to brain structure and function[41], suggestive of a bidirectional relation between sleep, BMI, and the brain.

Last, we observed a relation between sleep and depressive symptoms. A recent meta-analysis implicated both long and short sleep to be associated with increased risk of depression in adults[64]. Though the mechanisms underlying this association are not fully understood, daytime tiredness, resulting in increased negative events and emotions, has been shown to be predictive of poor outcome of depression. Next to this, sleep abnormalities relate also to low physical activity, which in turn modulates risk of depression. Importantly, sleep factors can predispose, precipitate, and perpetuate depression and in our multivariate model, we observed both neutral and positive associations between depression and unhealthy sleep behaviors, highlighting the complex relation between sleep and mental health.

Though we could establish phenotypic and genetic correlations between sleep duration and local cortical thickness in two independent samples, findings were inconsistent. In the HCP sample, but not in the eNKI sample, sleep duration was linked to thickness in the frontal areas. The important role of frontal cortex in sleep is previously well-documented. For example, sleep deprivation influences frontal executive functions in both healthy individuals and patients with insomnia disorder[65–67]. In addition, sleep deprivation leads to lower metabolism in the frontal cortex, while sleep recovery moderately restores frontal lobe functions[68]. Function abnormalities are also mirrored by abnormalities in macro-anatomical structure, where cortical thinning in bilateral precentral cortex and the superior/mid frontal cortex related to insomnia symptoms[69] and patients with insomnia disorder showed gray matter abnormalities in the frontal cortices[70,71]. On the other hand, phenotypic analyses in the eNKI sample demonstrated that sleep duration had a positive link with thickness in bilateral inferior temporal regions and right occipital cortex. Also function and structure of temporal and occipital areas has been associated with sleep patterns. For instance, older adults with short or long sleep duration had higher rates of cortical thinning in the frontal and temporal regions, as well as the inferior occipital gyrus[72] relative to older adults with normal sleep duration. Also, insomnia disorder has been related to functional abnormalities in the temporal and occipital areas, beside the frontal regions[73,74]. These activations have been associated with excessive hyperarousal, impaired alertness, auditory-related and vision-related inattention, and experiencing negative moods in such patients. Possible causes for divergence could be sample characteristics, as well as confounding effects. However, even when controlling for age, intelligence, BMI, and depression, findings remained dissimilar between samples. Only when evaluating spatial patterns of relationships between sleep

duration and cortical thickness, we observed cross-sample consistency, suggesting that the degree of impact of sleep duration on local brain structure varied across samples, but that the direction of the relation between sleep and cortical thickness was comparable across the cortex. Of note, though we observed diverging phenotypic relationships across samples, three out of four local relationships between sleep and cortical thickness were observed driven by additive genetic factors, suggestive of a system-level impact of sleep on brain structure, with modest but robust underlying local genetic associations. Nevertheless, local, univariate, associations between sleep and cortical thickness should be interpreted with caution, as they were not consistent across samples. It is of note that detrimental effects of both short and long sleep have been reported previously[46,75] and in the current study, the large majority of individuals (>65% for all samples) reported less than 7h of sleep whereas only a small proportion (<9%) slept 9 h or more in both samples. Indeed, post-hoc analysis indicated associations between local brain structure and sleep duration reflected patterns of short-to-normal sleep duration. Further studies in samples reporting long sleep duration are needed to evaluate the differential effects of short and long sleep duration on local brain structure.

Univariate relationships between sleep, brain structure, and behavior in two independent samples were further corroborated by our multivariate analysis. We could identify multivariate, latent, relationships between cortical thickness on the one hand and sleep, intelligence, BMI, and depression on the other one. In the first factor, reflecting a positive-negative axis of behavior, where sleep duration, together with intelligence, low BMI, and high sleep quality, showed a negative relation to thickness in the frontal and parietal areas, but a positive relation to the sensory-motor and parahippocampal areas. These latent relationships were robust across samples and driven by shared additive genetic effects. Behaviorally, the observed latent factor mirrors the previously reported positive-negative axis of behavior previously defined in a sub-sample of the HCP using functional connectivity. Here, the axis related to increased functional connectivity of the default mode network and negative associations in the sensorimotor networks[76]. We observed a second axis with a positive link between low sleep quality, high intelligence, high depression, and high BMI score and thickness of dorsolateral frontal cortex, accompanied by negative relation to thickness in sensorimotor areas. This axis related positively to intelligence and depression.

Our multivariate observations are broadly in line with our univariate results. Indeed, the first brain factor again highlights a negative relation between frontal thickness and sleep duration, whereas temporal-occipital regions show a positive relation with duration of sleep, reconciling divergent findings in the two independent samples. However, our latent model also provided a system-level perspective on the relation between sleep, behavior, and brain structure. Here, unimodal and heteromodal association cortices revealed an inverse relation to sleep and behavioral variability. A previous body of literature have put forward a so-called hierarchical model of brain function stretching from unimodal to transmodal cortices, enabling both externally, as well as internally oriented processing[77,78]. Indeed, it is likely that sleep, intelligence, BMI, and depression do not only relate to internally oriented processes, but also functional processes focused on the external world supported by the somatosensory cortices. For example, previous work has implicated sleep deprivation in sensorimotor coupling, reporting that sleep deprived individuals showed difficulties standing upright[79]. Likewise, memory consolidation processes during sleep have been linked to primary and secondary sensorimotor cortices. For example, in mice, inhibition of projecting axons from motor cortex to somatosensory cortex impaired sleep-dependent reactivation of sensorimotor neurons

and memory consolidation[80]. Similarly, other studies applying multivariate methods to understand the relation between system-level brain function and complex behavior also have implicated alterations of inter-network relationships between somatosensory and heteromodal association cortices in mental function and dysfunction[81,82]. It is possible that such disruptions are due to dissociable neurodevelopmental as well as genetic effects affecting the hierarchical interrelation of these brain systems[77]. Future research on the neurobiology of sleep requires to be conducted with functional and structural connectivity data enabling more direct analysis of the relation between system-level connectivity, sleep, and behavior.

In addition to providing evidence for a shared neurobiological basis of sleep, mental and physical health, we observed that, in line with previous literature[17,18,46], variance in global sleep quality and sleep duration was in part driven by additive genetic effects. A recent GWAS study using 446,118 adults from UK Biobank identified 78 loci, mainly PAX8 locus, for self-reported habitual sleep duration[46]. Moreover, Dashti et al. observed, similar to our observations, genetic overlap between sleep, markers of mental and physical health, as well as education attainment. It is likely that the observed genetic correlation within our sample between cortical thickness, intelligence, BMI, and sleep is due to mediated pleiotropy (a gene affects A, which affects B). Thus, it could be that a genetic mechanism affects gray matter macrostructure and associated function and, as a consequence, sleep duration. Alternatively, genetic variation might affect brain function, which in turn modulates both macroscale structure and sleep duration, or a genetic mechanism affects sleeping behaviors through non-brain processes and in turn affects brain function and structure. However, it is worthy to mention that our genetic correlations analysis does not provide causal mechanisms on the relation between brain structure, sleep, and behavior. Indeed, there is genetic evidence for a bidirectional relationship between sleep duration and schizophrenia[46], as well as smoking behavior[83], highlighting the complex interplay between and behavior. Next to this, though there is a negative genetic correlation between short sleep duration and long sleep duration[46], suggestive of shared biological mechanisms, it is possible that both relate to partly distinct underlying biological mechanisms. Further studies will be needed to investigate whether shorter and longer sleep duration differentially affect brain and behavior, and investigate the relation between sleep, health, and brain in longitudinal datasets with imaging and deep phenotyping to further disentangle causal relationships between sleep, brain structure, and function.

Our integrative perspective on sleep, behavior, and brain structure may be relevant for future work targeting the relation between sleep and neurodevelopment. For example, studies on development have indicated a close relation between sleep, behavioral problems, and school performance in children[53]. As childhood is an essential time for neurodevelopment[84,85] combining these two lines of research might help to understand how healthy and abnormal sleep patterns relate to neurodevelopment in youth. At the same time, sleep disturbances have been related to neurodegenerative conditions, and may drive early-onset pathogenesis. For example, sleep disruption has been observed to upregulate neuronal activity, which increases the production of amyloid-beta proteins resulting in exacerbated tau pathology in various mouse models[86] and sleep disturbances in ageing might directly influence synaptic homeostasis and cognitive function. By providing system-level evidence integrating cortical thickness with sleep and behavior, follow-up research could further disseminate how brain anatomy relates to sleep and general functioning during development and ageing and identify functional and structural mechanisms that explain the interrelation between sleep, development and ageing.

**Table 4 Behavioral characteristics of the HCP unrelated sample.**

| Measure | n | Mean ± SD (range) |
|---|---|---|
| Males/females | 196/228 | – |
| Age | 424 | 28.6 ± 3.7 (22–36) |
| Sleep duration (hours) | 424 | 6.8 ± 1.2 (2.5–10) |
| Total sleep quality | 424 | 4.9 ± 2.8 (0–15) |
| BMI | 424 | 26.6 ± 5.3 (16.7–44.5) |
| Intelligence (total cognitive score) | 418 | 121.5 ± 14.7 (84.6–153.4) |
| Depression (DSM-scale) | 419 | 54.1 ± 6.1 (50–87) |

**Table 5 Behavioral characteristics of the complete HCP sample including twins and siblings.**

| Measure | n | Mean ± SD (range) |
|---|---|---|
| Males/females | 507/606 | – |
| Age | 1113 | 28.8 ± 3.7 (22–37) |
| Sleep duration (hours) | 1113 | 6.8 ± 1.1 (2.5–12) |
| Total sleep quality | 1113 | 4.8 ± 2.8 (0–19) |
| BMI | 1112 | 26.5 ± 5.2 (16.5–47.8) |
| Intelligence (total cognitive score) | 1096 | 121.8 ± 14.6 (84.6–153.4) |
| Depression (DSM scale) | 1105 | 53.9 ± 5.7 (50–87) |

**Table 6 Behavioral characteristics of the eNKI sample.**

| Measure | n | Mean ± SD (range) |
|---|---|---|
| Males/females | 296/487 | – |
| Age | 783 | 41.2 ± 20.3 (12–85) |
| Sleep duration (hours) | 783 | 6.9 ± 1.3 (3–12) |
| Total sleep quality | 783 | 4.6 ± 3.2 (0–17) |
| BMI | 757 | 27.1 ± 5.9 (15.7–50.0) |
| Intelligence (WASI) | 783 | 101.9 ± 13.3 (65–141) |
| Depression (BDI) | 782 | 4.21 ± 6.3 (0–40) |

Notably, we refrain from interpreting environmental correlations, as the environmental component includes environmental factors, but also measurement errors. In a previous work[87], we have shown that a model based on genetic and environmental factors only was more parsimonious compared to a model including common household effects in extended family samples such as the HCP sample. Nevertheless, as individual variance in sleep duration and quality was only in part explained by genetic factors, future longitudinal models might help uncover relevant familial and non-familial environmental effects relating sleep to mental and physical health.

Taken together, our study on the interrelation between sleep, mental, and physical health and brain structure was made possible by the open HCP and eNKI neuroimaging repositories. These initiatives offer cognitive neuroimaging communities a unique access to large datasets for the investigation of the brain basis of individual difference. The use of multiple datasets has enabled us to highlight variability across samples, and allowed us to preform validation experiments to verify stability of our observations. Given that reproducibility is increasingly important nowadays, our study illustrates the advantages of open data to increase our understanding of complex traits.

## Methods

**Participants and study design: human connectome project.** We studied the publicly available Human Connectome Project dataset (HCP; http://www.humanconnectome.org/), which included data from 1206 individuals (656 females), 298 monozygotic twins (MZ), 188 dizygotic twins (DZ), and 720 singletons, with mean age 28.8 years (SD = 3.7, range = 22–37). Participants for whom the images and data had been released (humanconnectome.org) after passing the HCP quality control and assurance standards were included. The full set of inclusion and exclusion criteria are described elsewhere[88]. All participants signed an informed consent document at the beginning of day 1 of testing.

For our phenotypic analyses, we selected an unrelated subsample with complete behavioral data (n = 457). After removing individuals with missing structural imaging our sample for phenotypic correlations consisted of 424 (228 females) individuals with mean age of 28.6 years (SD = 3.7, range = 22–36), see further Table 4. For our twin-based genetic analyses, we used the complete sample of individuals with complete structural imaging for structural gray matter and behavioral data for sleep genetic correlation analyses including 1105 individuals (599 females), 285 MZ twins, 170 DZ twins, and 650 singletons, with mean age 28.8 years (SD = 3.7, range = 22–37), see further Table 5. Environmental correlations were also derived in this sample as a by-product of analysis of genetic correlation analysis.

**Structural imaging processing: human connectome project.** MRI protocols of the HCP are previously described[89,90]. In particular, the applied pipeline to obtain the FreeSurfer-segm entation is described earlier[89] and is recommended for the HCP data. The pre-processing steps included co-registration of T1 and T2 images, B1 (bias field) correction, and segmentation and surface reconstruction using FreeSurfer version 5.3-HCP to estimate cortical thickness[89].

**Participants and study design: eNKI sample.** To evaluate the cross-sample reproducibility of observations, we additionally investigated correspondence between sleep and cortical brain structure in the enhanced Nathan Kline Institute-Rockland Sample (NKI). The sample was made available by the Nathan-Kline Institute (NKY, NY, USA), as part of the 'enhanced NKI-Rockland sample' (https://www.ncbi.nlm.nih.gov/pmc/articles/PMC3472598/). All approvals regarding human subjects' studies were sought following NKI procedures. Images were acquired from the International Neuroimaging Data Sharing Initiative (INDI) online database https://fcon_1000.projects.nitrc.org/indi/enhanced/studies.html.

For our phenotypic analyses, we selected individuals with complete sleep and imaging data. Our sample for phenotypic correlations consisted of 783 (487 females) individuals with mean age of 41.2 years (SD = 20.3, range = 12–85). See Table 6 for demographic characteristics.

**Structural imaging processing: NKI Rockland sample.** 3D magnetization-prepared rapid gradient-echo imaging (3D MP-RAGE) structural images[91] were acquired using a 3.0 T Siemens Trio scanner with TR = 2500 ms, TE = 3.5 ms, Bandwidth = 190 Hz/Px, field of view = 256 × 256 mm, flip angle = 8°, voxel size = 1.0 × 1.0 × 1.0 mm. More details on image acquisition are available at https://fcon_1000.projects.nitrc.org/indi/enhanced/studies.html. All T1 images were visually inspected to ensure the absence of gross artefacts and subsequently pre-processed using the FreeSurfer software library (http://surfer.nmr.mgh.harvard.edu/) Version 5.3.0[92].

**Parcellation-summaries of cortical thickness.** We used a parcellation scheme[50] based on the combination of a local gradient approach and a global similarity approach using a gradient-weighted Markov Random models. The parcellation has been comprehensively evaluated with regards to stability and convergence with histological mapping and alternative parcellations. In the context of the current study, we focused on the granularity of 200 parcels. In order to improve signal-to-noise and improve analysis speed, we opted to average unsmoothed structural data within each parcel and cortical thickness of each region of interest (ROI) was estimated as the trimmed mean (10 percent trim).

**Selection of behavioral markers based on HCP phenotypic traits.** First, to constrain analyses, we selected primary markers for cognition, mental and physical health based on the relation of sleep to these traits in HCP. The selected traits include 38 emotional, cognitive, NEO-FFI personality, as well as the 7 PSQI sleep markers for reference, based on the unrestricted phenotypic data, as well as 46 mental and physical health markers based on the restricted phenotypic data. For more information on available phenotypes, see: https://wiki.humanconnectome.org/display/PublicData.

**Behavioral markers: HCP.** Inter-individual difference in sleep quality was derived from information of the self-reported Pittsburg Sleep Questionnaire (PSQI)[11], which is a common measure of sleep quality with significant item-level reliability and validity.

For markers of life function, we used BMI (703 × weight/(height)²) and the ASR depression DSM-oriented scale for ages 18–59[49] (https://aseba.org/). The ASR is a

self-administered test examining diverse aspects of adaptive functioning and problems. Scales are based on 2020 referred adults and normed on 1767 non-referred adults. The test-retest reliability of the ASR was supported by 1-week test-retest that were all above 0.71. The ASR also has good internal consistency (0.83), and in the current study we focused on depression sub-score.

As a proxy for intelligence, we used the NIH Toolbox Cognition[47], 'total composite score'. The Cognitive Function Composite score is derived by averaging the normalized scores of each of the Fluid and Crystallized cognition measures, then deriving scale scores based on this new distribution. Higher scores indicate higher levels of cognitive functioning. Participant score is normed to those in the entire NIH Toolbox Normative Sample (18 and older), regardless of age or any other variable, where a score of 100 indicates performance that was at the national average and a score of 115 or 85, indicates performance 1 SD above or below the national average.

**Behavioral markers: NKI**. Sleep markers were derived from the Pittsburg Sleep Questionnaire (see further the section on this question in the HCP sample).

Depression was measured using the Beck Depression Inventory (BDI–II). The BDI-II is a 21-item self-report questionnaire assessing the current severity of depression symptoms in adolescents and adults (ages 13 and up). It is not designed to serve as an instrument of diagnosis, but rather to identify the presence and severity of symptoms consistent with the criteria of the DSM-IV. Questions assess the typical symptoms of depression such as mood, pessimism, sense of failure, self-dissatisfaction, guilt, punishment, self-dislike, self-accusation, suicidal ideas, crying, irritability, social withdrawal, insomnia, fatigue, appetite, and loss of libido. Participants are asked to pick a statement on a 4-point scale that best describes the way they have been feeling during the past two weeks[93]. Body-mass-index was calculated using weight and height. These vitals are obtained and recorded by study staff. Height was recorded in centimeters. Weight was recorded in kilograms. Body Mass Index (BMI) was automatically calculated.

Intelligence was measured using the Wechsler Abbreviated Scale of Intelligence (WASI-II). The WASI is a general intelligence, or IQ test designed to assess specific and overall cognitive capabilities and is individually administered to children, adolescents and adults (ages 6-89). It is a battery of four subtests: Vocabulary (31-item), Block Design (13-item), Similarities (24-item) and Matrix Reasoning (30-item). In addition to assessing general, or Full Scale, intelligence, the WASI is also designed to provide estimates of Verbal and Performance intelligence consistent with other Wechsler tests. Specifically, the four subtests comprise the full scale and yield the Full-Scale IQ (FSIQ-4). The Vocabulary and Similarities subtests are combined to form the Verbal Scale and yield a Verbal IQ (VIQ) score, and the Block Design and Matrix Reasoning subtests form the Performance Scale and yield a Performance IQ (PIQ) score[48].

### Statistics and reproducibility

*Phenotypic analysis*. For our phenotypic analysis in the HCP sample, we selected an unrelated subsample to overcome possible bias due to genetic similarity of individuals. In eNKI the complete sample with available data was used. To assess phenotypic relationships between sleep parameters and behavior/brain structure, we used Spearman's correlation test to account for outliers, while controlling for age, sex, age × sex interaction, $age^2$, $age^2$ × sex interaction. In our structural whole-brain analysis, we additionally controlled for global thickness. Findings were similar when additionally controlling for depression, BMI or intelligence. We controlled for multiple comparisons at FDR $q < 0.05$, per analysis step of univariate behavior and univariate brain analysis, and report FDR q thresholds for reference. We used the Robust Correlation Toolbox for Matlab to define confidence intervals in our post-hoc phenotypic correlations[94].

*Heritability and genetic correlation analysis*. To investigate the heritability and genetic correlation of sleep parameters and brain structure, we analyzed sleep parameters and 200 parcels of cortical thickness of each subject in a twin-based heritability analysis. As previously described[95], the quantitative genetic analyses were conducted using Sequential Oligogenic Linkage Analysis Routines (SOLAR)[96]. SOLAR uses maximum likelihood variance-decomposition methods to evaluate the relative importance of familial and environmental influences on a phenotype by modeling the covariance among family members as a function of genetic proximity. Coefficient of relationship (genetic proximity) between individuals in the HCP sample was computed using the KING method in the openly available genotyped data of HCP. The method is described in detail before[97] and evaluated in the context of the current sample as described previously[98]. This approach can handle pedigrees of arbitrary size and complexity and thus is optimally efficient with regard to extracting maximal genetic information. To ensure that neuroimaging traits, parcels of cortical thickness, conform to the assumptions of normality, an inverse normal transformation was applied[95].

Heritability ($h^2$) represents the portion of the phenotypic variance ($\sigma_p^2$) accounted for by the total additive genetic variance ($\sigma_g^2$), i.e., $h^2 = \sigma_g^2/\sigma_p^2$. Phenotypes exhibiting stronger covariances between genetically more similar individuals than between genetically less similar individuals have higher heritability. Within SOLAR, this is assessed by contrasting the observed covariance

matrices for a phenotypic (neuroimaging or behavioral) measure with the structure of the covariance matrix predicted by kinship. Heritability analyses were conducted with simultaneous estimation for the effects of potential covariates. For this study, we included covariates of age, sex, age × sex interaction, $age^2$, $age^2$ × sex interaction. When investigating cortical thickness, we additionally controlled for global thickness effects, as well as depression score, BMI, and intelligence in post-hoc tests. Heritability estimates were corrected for multiple comparisons at FDR $q < 0.05$, controlling for the number of parcels in case of analysis of brain structure.

We performed genetic correlation analysis to determine if variations in sleep and cortical thickness were influenced by the same genetic factors. Specifically, bivariate polygenic analyses were conducted to estimate genetic ($\rho_g$) and environmental ($\rho_e$) correlations, based on the phenotypic correlation ($\rho_p$), between brain structure and sleep with the following formula:
$\rho_p = \rho_g\sqrt{(h_1^2 h_2^2)} + \rho_e\sqrt{[(1 - h_1^2)(1 - h_2^2)]}$, where $h_1^2$ and $h_2^2$ are the heritability's of the parcel-based cortical thickness and the sleep parameters. The significance of these correlations was tested by comparing the log likelihood for two restricted models (with either $\rho_g$ or $\rho_e$ constrained to be equal to 0) against the log likelihood for the model in which these parameters were estimated. A significant genetic correlation (using a FDR $q < 0.05$) is evidence suggesting that both phenotypes are influenced by a gene or set of genes[99].

**Partial least squares**. PLS is a multivariate data-driven statistical technique that aims to maximize the covariance between two matrices by deriving latent components (LCs), which are optimal linear combinations of the original matrices[100,101]. We applied PLS to the cortical thickness and sleep, BMI, depression and IQ measures of all participants. In short, PLS performs data normalization, cross-covariance, and singular value decomposition. Following, brain and behavioral scores are created and permutation testing is performed to assess significance of each latent factor solution. Last, bootstrapping is performed to test the stability of the brain saliencies.

Each LC has a distinct cortical thickness pattern (called brain saliences) and a distinct behavioral profile (called behavioral saliences). By linearly projecting the cortical thickness and behavioral measures of each participant onto their corresponding saliences, we obtain individual-specific brain and behavioral composite scores for each LC. PLS seeks to find saliences that maximize across-participant covariance between the brain and behavioral composite scores. The number of significant LCs was determined by a permutation (1000 permutations). The p-values (from the permutation test) for the LCs were corrected for multiple comparisons using a false discovery rate (FDR) of $p < 0.05$. For the brain saliencies, though all regions contributed to the latent brain score, we highlighted regions with a bootstrap ratio > 2, approximately $p < 0.05$. Findings where summarized at the level of macroscale function networks[102], by averaging the BSR score per network, as well as summarizing the relative contribution of each functional network to positive (BSR > 2), as well as negative (BSR < −2) relations. Here we controlled for the size of the network.

**Functional decoding**. All significant parcels were functionally characterized, using the Behavioral Domain meta-data from the BrainMap database using forward inference (www.brainmap.org)[103,104]. To do so, volumetric counterparts of the surface-based parcels were identified. In particular, we identified those meta-data labels (describing the computed contrast [behavioral domain]) that were significantly more likely than chance to result in activation of a given parcel[105–107]. That is, functions were attributed to the identified effects by quantitatively determining which types of experiments are associated with activation in the respective parcellation region. Significance was established using a binomial test ($p < 0.05$, corrected for multiple comparisons using false discovery rate (FDR)) (Supplementary Fig. 6).

**Reporting summary**. Further information on research design is available in the Nature Research Reporting Summary linked to this article.

## Data availability

All data, analyzed in this manuscript, were obtained from the open-access HCP young adult sample (http://www.humanconnectome.org/)[90] and enhanced NKI-Rockland sample (https://www.ncbi.nlm.nih.gov/pmc/articles/PMC3472598/)[51]. Brain images were acquired from the International Neuroimaging Data Sharing Initiative (INDI) online database http://fcon_1000.projects.nitrc.org/indi/enhanced/studies.html. The raw data may not be shared by third parties due to ethics requirements, but can be downloaded directly via the above weblinks. Spearman correlations and confidence intervals were computed using the Robust Correlation toolbox https://github.com/CPernet/robustcorrtool[94].

Genetic analyses were performed using solar eclipse 8.4.0 (http://www.solar-eclipse-genetics.org), and data on the KING pedigree analysis is available here: https://www.nitrc.org/projects/se_linux/[96,98]. We performed partial least square analysis using https://miplab.epfl.ch/index.php/software/PLS.[100,101]. BrainMap analysis were performed using http://www.brainmap.org[103,104].

## Code availability

Main analysis scripts and genetic correlation tables are available at https://github.com/sofievalk/projects/tree/master/Tahmasian_Sleep.

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

## Acknowledgements

We would like to thank the various contributors to the open access databases that our data was downloaded from. Specifically; HCP data were provided by the Human Connectome Project, Washington University, the University of Minnesota, and Oxford University Consortium (Principal Investigators: David Van Essen and Kamil Ugurbil;1U54MH091657) funded by the 16 NIH Institutes and Centers that support the NIH Blueprint for Neuroscience Research; and by the McDonnell Center for Systems Neuroscience at Washington University. For enhanced NKI, we would like to thank the principal support for the enhanced NKI-RS project is provided by the NIMH BRAINS R01MH094639-01 (PI Milham). Funding for key personnel was also provided in part by the New York State Office of Mental Health and Research Foundation for Mental Hygiene. Funding for the decompression and augmentation of administrative and phenotypic protocols provided by a grant from the Child Mind Institute (1FDN2012-1). Additional personnel support provided by the Center for the Developing Brain at the Child Mind Institute, as well as NIMH R01MH081218, R01MH083246, and R21MH084126. Project support also provided by the NKI Center for Advanced Brain Imaging (CABI), the Brain Research Foundation (Chicago, IL), and the Stavros Niarchos Foundation. This study was supported by the Deutsche Forschungsgemeinschaft (DFG, EI 816/21-1), the National Institute of Mental Health (R01-MH074457), the Helmholtz Portfolio Theme "Supercomputing and Modeling for the Human Brain" and the

European Union's Horizon 2020 Research and Innovation Program under Grant Agreement No. 785907 (HBP SGA2). B.T.T.Y. is supported by the Singapore National Research Foundation (NRF) Fellowship (Class of 2017). Any opinions, findings and conclusions or recommendations expressed in this material are those of the author(s) and do not reflect the views of the Singapore NRF. This study was also supported by NIH grants R01EB015611 and S10OD023696 to P.K.

## Author contributions

S.L.V., M.T. and F.S. designed the experiments and wrote the paper. S.L.V. performed analyses. F.H. processed imaging data. J.A.C. performed the functional decoding analysis and approved the paper. P.K. created the twin-pedigree for genetic analysis. H.K., M.Z., S.K., B.T.T.Y. and S.B.E. revised and approved the paper.

## Competing interests

The authors declare no competing interests.
