## [Peer Review File · Communications Biology]

Reviewers' comments:

Reviewer #1 (Remarks to the Author):

The authors report analyses linking sleep, BMI, depression, intelligence and brain structure in the HCP and eNKI studies. I applaud the authors for reporting on both datasets, and for the substantial amount of analysis included, however, I have a number of concerns about the manuscript.

Major

(1) Sleep duration is considered as a linear variable. However, the detrimental effects of both short and long sleep on health are well-documented. Therefore, the analyses as they stand may not be sensitive to associations with sleep duration.

(2) The eNKI study has a wide age range – spanning from 12 to 85. Recommended amounts of sleep vary across this age range. While the authors covary for age, and age x age, further exploration of age is warranted.

(3) The interpretation and take-home message of the salience analyses is unclear

(4) No hypotheses are provided

Minor

Abstract

- Age range of participants should be added

Introduction

- "The National Sleep Foundation suggests seven to nine hours of sleep per night for young adults" – please expand to explain the recommendations for the age range of participants included in the study

- "whereas depressive symptoms, physical illness, and fatigue were reported as risk factors for both poor sleep quality and short sleep duration" the wording seems to imply that the association is uni-directional; please make clear that bi-directional relationships are possible

- "Moreover, white matter integrity underlying prefrontal areas has been associated with sleep duration and sleep quality" – in addition to the original study cited, it may be worthwhile to cite review articles in this field for insomnia (e.g. Spiegelhalder et al 2013 – neuroimaging studies in insomnia; Sexton et al 2019 – connections between insomnia and cognitive aging)

Results

- Can intro statements on characterisation of sleep traits be added – what was the average and range of sleep duration, PSQI? How many displayed short or long sleep, or met caseness for PSQI?

Characterisation of other variables is also needed – how severe were depressive symptoms etc.

- "Evaluating the relation between sleep and our selected markers in eNKI, in addition to HCP, we observed that sleep duration had a consistent negative phenotypic relation to both BMI and depression, and a positive relation to IQ." – can the table where these results are detailed be referenced

- "Behaviorally, we observed a strong negative correlation (Spearman $r=-0.51$ [-0.59 -0.44], $p<0.001$) between global sleep quality and sleep duration (Fig. 1A)." – Fig 1A doesn't illustrate the correlation, rather displays histograms, can the correlation be shown?

Methods

- "We included for whom the scans" should be "we included participants for whom the scans"?

- How was sleep duration considered? Short and long sleep associated with detrimental effects

Discussion

- Limitations of using general cognition as marker of intelligence across the age range included should be discussed

- "In line with our observations, Lim and Dinges report a relation between complex attention on working memory." – a reference is needed here. Also, it's unclear if there's something missing, is the report in relation to sleep deprivation? And it's arguable if findings on specific domains (and not others) is in line with associations with IQ?

Fig 2

- A, D, B, I, Q should be explained in abbreviations. Why is A sleep duration?

Supplementary Material

- Table S2 and S3, can the significance threshold be repeated in table legend please?

There are quite a few typos, (For example, "sleep deprivation influences frontal executive functions in both health individuals and patients. In addition, sleep deprivation disrupts leads to lower metabolism in the frontal cortex) and the manuscript could do with a final proof read for such errors

Reviewers' comments:

Reviewer #1

The authors report analyses linking sleep, BMI, depression, intelligence and brain structure in the HCP and eNKI studies. I applaud the authors for reporting on both datasets, and for the substantial amount of analysis included, however, I have a number of concerns about the manuscript.

We would like to thank the reviewer for the constructive assessment and the recommendations, which we were happy to incorporate in the revised version.

(1) Sleep duration is considered as a linear variable. However, the detrimental effects of both short and long sleep on health are well-documented. Therefore, the analyses as they stand may not be sensitive to associations with sleep duration.

We thank the reviewer for pointing this out. Indeed, both extreme short and long sleep duration can have adverse effects on health¹⁻³ and various previous studies have taken a categorical approach (short vs. long) for sleep duration^{4,5}. In the current datasets there are relatively few participants with longer than healthy (7-9 hours⁶) sleep and as our main focus was to study how the duration of sleep linearly related to local brain structure, and markers of health, similar to previous studies⁶⁻⁸, we chose not to focus specifically on detrimental effects of deviation from normal sleep amount (e.g. shorter than average or longer than average sleep. Notably, in the unrelated HCP sample only few (12/424) individuals reported to sleep 9 hours or longer, whereas 133 individuals reported to sleep between 7 and 9h and 279 individuals reported to sleep 7 hours or shorter. In the eNKI sample, (155/783) individuals reported normal sleep, (559/783) reported short sleep, and (69/783) reported long sleep. As the distribution of both samples is such as the majority of individuals report less than normal sleep, further analysis comparing short-vs-normal, and long-vs-normal sleep are challenging in these databases. Nevertheless, according to this suggestion, we now highlight the distribution of short vs. long sleepers in both samples in the results and discussion to further put reported observations in perspective. We have reported the distributions of short-normal-long sleepers in the sample description in the Table S5.

	HCP unrelated	HCP complete	eNKI
Short	279/424	750/1113	559/783
Normal	133/424	328/1113	155/783
Long	12/424	35/1113	69/783

Table S5. Number of short, normal and long sleepers in the HCP and eNKI sample. Here groups are defined similarly to previous study⁴. Short sleepers: 7h or shorter; normal sleepers: between 7 and 9h of sleep, long sleepers: 9h or longer.

However, in accordance with the reviewers suggestions, we now performed post-hoc analysis on the relationship between short-to-normal (<9h) sleep duration and local brain structure as well as normal-to-long (≥7h) and local brain structure. Results are displayed below, as well as included in Figure S2.

Figure S2. Relationship between short and long sleep duration and brain structure. A) Linear relationship between short to normal sleep (<9h) and local cortical thickness in the unrelated HCP subsample; B) Linear relationship between normal to long sleep (≥ 7 h) and local cortical thickness in the unrelated HCP subsample; C) Linear relationship between short to normal sleep (<9h) and local cortical thickness in the eNKI sample; D) Linear relationship between normal to long sleep (≥ 7 h) and local cortical thickness in the eNKI sample.

In the result section in page 10:

“In both samples, most individuals (>65%) slept fewer than the recommended 7-9 hours (Table S5) and only a small proportion of both samples (<9%) slept 9 hours or more. Post-hoc analysis evaluating the linear relationship between short and long sleep duration and local brain structure replicated overall effects between sleep duration and local brain structure in individuals who report to sleep less than 9h per night (short-to-normal sleep duration), but not in individuals sleeping more than 7h per night (normal-to-long sleep duration) (Figure S2).”

And discussed the relation between sleep duration, the distribution of long and short sleep in the current samples in the discussion (page 21):

“Of note, in the current study, we studied the linear relationship between sleep, cortical thickness, and mental and physical health. Importantly, detrimental effects of both short and long sleep have been reported previously^{4,5}. In the current study, the large majority of individuals (>65% for all samples) reported less than 7h of sleep, and only a small proportion (<9%) slept 9

hours or more in both samples studied. Indeed, post hoc analysis indicates associations between local brain structure and sleep duration reflect patterns of short-to-normal sleep duration. Further studies in samples reporting long sleep duration are needed to evaluate the differential effects of short and long sleep duration on local brain structure.”

(2) The eNKI study has a wide age range – spanning from 12 to 85. Recommended amounts of sleep vary across this age range. While the authors covary for age, and age x age, further exploration of age is warranted.

We thank the reviewer for this insightful remark. Indeed, recommended sleep duration is different across various age ranges based on National Sleep Foundation. In the current study, we performed our main analysis in two samples with various age ranges (page 4: young adults in HCP (22-37yrs) and lifespan sample in eNKI (12-85yrs)) in order to evaluate “generalizable effects” of sleep duration and quality on local brain structure and markers of health. Thus, we included the complete sample of eNKI rather than an age-matched sub-sample of eNKI. We have included the following sentence in the introduction to further motivate this decision on page 4:

“The HCP sample consists of young adults only, whereas the eNKI sample consists of both children, younger and older adults, enabling us to evaluate the generalizability of the interrelation of sleep, health and local brain structure”

In order to further evaluate the relationship between amount of sleep and age, we have additionally performed various follow-up analyses in the eNKI sample. First, we assessed the relationship between sleep duration in youth (under 18 yrs., n=100), adult (18-50 yrs., n=393), and elderly (50+ yrs., n=290) populations. Following, we assessed whether there are differential effects of age vs. brain across these groups. In none of these analyses, we observed significant (FDRq<0.05) associations between sleep duration and local brain structure. We now report the results of these analyses in the Supplementary results (Table S6, Figure S3):

eNKI	Sleep duration $m(sd)$ min - max
Youth (12-17)	7.5(1.5) 4-12
Adult (18-49)	6.8(1.3) 3-11
Elderly (50-85)	6.7(1.2) 3-10

Table S6. Sleep duration based on three age groups in the eNKI sample.

Figure S3. Age-specific modulation of the relation between sleep duration and cortical thickness. In the eNKI sample, we split the sample in youths (12-17), adults (18-49), and elderly individuals (50-85) and evaluated the relationship between thickness and sleep duration in each of these age-group separately as well as compared whether effects of sleep duration on brain structure were different between groups. Findings at $FDR_{q<0.05}$ have a black outline, and trends at $p<0.01$ are reported at 60% transparency.

We have included the following sentence in page 10:

“As the eNKI sample had a broad age range from 12 to 85 years of age, we performed several follow up analysis to evaluate the relationship between sleep duration and brain structure in

youths, adults and elderly populations (Table S6). Here, would could not observe differential sleep duration effects in each sub-group, as well as differences between age-groups at $FDRq < 0.05$ (Figure S3)."

(3) *The interpretation and take-home message of the salience analyses is unclear.*

We agree that we need to further clarify the interpretation of the latent component analysis. As these analyses are exploratory, we are cautious in interpretation of these results. However, in combination with direct correlational analysis between sleep pattern and mental and physical health, we showed, using data-driven approaches, that there is a shared variance in nocturnal behaviors and mental and physical health, and that these latent factors relate to variation in local brain structure. This suggests that associations between sleep duration and local brain structure are not specific, i.e. other non-nocturnal factors also contribute to the associations between sleep and brain. Further studies are needed to evaluate possible causal mechanisms between associations of sleep, mental and physical health, and local brain structure.

Latent component analysis is a data-driven method that enables to identify latent relationships between (in our case) markers of behavior and of local cortical thickness. Based on the observed phenotypic associations between sleep, depression, BMI and intelligence, and, in part, overlapping association with local brain structure, we explored whether there would also be an underlying shared relationship between these behavioral markers and brain structure.

In these exploratory analyses, we identified two latent components in the eNKI, which were replicated in the HCP dataset. The first component related most strongly to intelligence, and in lesser extend *long* sleep duration and *positive* sleep quality (inverse), and negatively on BMI. This suggests that the shared variance of intelligence and sleep duration has a multivariate relationship with sensory-motor cortices (positive association), and fronto-parietal and default mode regions (negative association). The second component was different, in that it explained shared variance of intelligence, *lack* of sleep quality, increased BMI and increased depression score. This latent factor showed a positive association with sensory-motor regions and a negative association with lateral frontal regions.

Our exploratory latent component analysis thus revealed two components underlying shared variance in reported sleep hygiene, intelligence, BMI and depression, each with a unique association with local brain structure. The first component is interpreted as a positive-negative axis of behavior, similar to the component reported previously⁹. The second component showed high loadings of depression, low sleep quality and intelligence, possibly related to ‘ruminative or internalizing’ behavior.

We have now added a further section in the discussion in page 25:

“Observations of the exploratory latent factor approach indicate that associations between sleep and local brain structure are likely not specifically driven by nocturnal behavior. Indeed, shared variance between sleep, mental and physical health relates to variance in local brain structure. Further work is needed to understand the causal relationships between nocturnal and non-nocturnal phenotypes and brain structure.”

Moreover, to avoid over-interpretation, we have excluded the supplementary post-hoc analysis on the behavioral association between saliencies and behavioral markers in the HCP sample.

(4) *No hypotheses are provided.*

We thank the reviewer for this comment. Indeed, we did not provide clear a priori hypothesis for the current work, as it was data-driven in nature. However, as stated in the introduction, we expected to find a shared variance of sleep and markers of mental and physical health and that individual difference in sleep would relate to individual difference in local brain structure. We also expected that this shared phenotypic variance could be due to shared genetic factors. We have now stated these expectations and motivations for the study more explicitly in the introduction in page 5:

“Based on previous knowledge, we expect to find a phenotypic relationship between sleep duration/quality and markers of mental and physical health. Moreover, we expect to observe a phenotypic relation between sleep and local grey matter structure.”

(5) Age range of participants should be added.

According to this suggestion, we have now added the age range in the abstract.

(6) “The National Sleep Foundation suggests seven to nine hours of sleep per night for young adults” – please expand to explain the recommendations for the age range of participants included in the study.

As suggested, we have revised the introduction to make it clearer in page 3:

“The National Sleep Foundation suggests 7-9 hours of sleep per night for adults (18-64) and 7-8 hours for older adults (65+). For school aged children (6-13) this is 9-11 hours, and for teenagers 8-10 hours of sleep.”

(7) “whereas depressive symptoms, physical illness, and fatigue were reported as risk factors for both poor sleep quality and short sleep duration” the wording seems to imply that the association is unidirectional; please make clear that bi-directional relationships are possible

We agree that those associations are bi-directional. Hence, we revised the text in page 3:

“Moreover, sleep can have a bidirectional relation with health. Not only sleep disturbance is linked with hypertension, diabetes, and obesity^{10,11}, but also depressive symptoms, physical illness, and fatigue were reported as associated factors for both poor sleep quality and abnormal sleep duration^{12,13}.”

(8) “Moreover, white matter integrity underlying prefrontal areas has been associated with sleep duration and sleep quality” – in addition to the original study cited, it may be worthwhile to cite review articles in this field for insomnia (e.g. Spiegelhalder et al 2013 – neuroimaging studies in insomnia; Sexton et al 2019 – connections between insomnia and cognitive aging).

We thank the reviewer for the helpful literature suggestions and have added the references on page 4.

(9) Can intro statements on characterization of sleep traits be added – what was the average and range of sleep duration, PSQI? How many displayed short or long sleep, or met caseness for PSQI? Characterization of other variables is also needed – how severe were depressive symptoms etc.

We thank the reviewer for the comments. Indeed, in the methods and supplementary tables, we have reported further characteristics of the sample. We have additionally added the information of the Table 4-6 in text for further convenience.

(10) “Evaluating the relation between sleep and our selected markers in eNKI, in addition to HCP, we observed that sleep duration had a consistent negative phenotypic relation to both BMI and depression, and a positive relation to IQ.” – can the table where these results are detailed be referenced.

Many thanks for this suggestion. A “Table 1” has now been added on page 8.

(11) “Behaviorally, we observed a strong negative correlation (Spearman $r=-0.51$ [-0.59 -0.44], $p<0.001$) between global sleep quality and sleep duration (Fig. 1A).” – Fig 1A doesn’t illustrate the correlation, rather displays histograms, can the correlation be shown?

We thank the reviewer for the comment and have now added the negative correlation between global sleep quality and sleep duration for HCP to Figure 1A as well as for eNKI to figure 1D.

(12) “We included for whom the scans” should be “we included participants for whom the scans”?

Many thanks for noticing the lapsus calami. This sentence has now been replaced in page 27.

(13) How was sleep duration considered? Short and long sleep associated with detrimental effects.

We thank the reviewer for this comment. Please see our response to Q1.

(14) Limitations of using general cognition as marker of intelligence across the age range included should be discussed.

We thank the reviewer for this insightful remark. Indeed, general cognition as a marker for intelligence across broad age ranges has been debated in the literature, as various sub-domains of cognitive functioning show differential age-related development and decline^{14,15}. In the HCP sample, the only general measure of cognitive aptitude was the ‘Cognition Total Composite’. This is a composite from measures of fluid (Flanker, Dimensional Change Card Sort, Picture Sequence Memory, List Sorting and Pattern Comparison) and crystallized cognition (Picture Vocabulary and Reading Tests)¹⁶. Observations were replicated in a large sample with broad age range, with difference in the available behavioral phenotypes. Here the marker closest to general cognitive skill was intelligence. Thus, we used the Wechsler Abbreviated Scale of Intelligence (WASI-II). The intelligence test is designed to assess specific and overall cognitive capabilities and is individually administered to children, adolescents and adults (ages 6-89). Importantly, previous studies also highlight variability between sub-domains of cognition, such as fluid and crystallized intelligence, in relation to genes and local brain structure^{17,18} and further study is needed to assess the relationship between sleep and domain-specific cognitive functioning across the lifespan. We have now added this important point to the discussion page 19:

“We observed a positive phenotypic relationship between amount of sleep and domain-general cognitive skill in the HCP sample, consisting of young healthy adults, as well as in the eNKI sample which included a broad age-range with children, adults, and elderly. Of note, though the measurement of domain-general cognition was not consistent across two samples as we used NIH Toolbox Cognition in HCP and the WASI-II in eNKI both tests have been validated for different age-ranges¹⁶. Further studies are needed to uncover the causal and longitudinal relationship between sleep, domain general, and domain specific cognitive skill across the life-span.”

(15) “In line with our observations, Lim and Dinges report a relation between complex attention on working memory.” – a reference is needed here. Also, it’s unclear if there’s something missing, is the report in relation to sleep deprivation? And it’s arguable if findings on specific domains (and not others) is in line with associations with IQ?

We have rewritten this sentence for clarification and added the reference on page 19:

“In line with our observations, Lim and Dinges show that short term sleep deprivation has a deleterious effect on a broad range of cognitive domains¹⁹.”

(16) Fig 2 - A, D, B, I, Q should be explained in abbreviations. Why is A sleep duration?

We thank the reviewer for the comment and have updated the abbreviations in the figure 2 and explanations of abbreviations in the legend of the figure.

Figure 2. Two latent dimensions of cortical macrostructure and components of sleep, mental and physical health.
A). Bootstrap ratio of the first brain saliency that showed significant robustness, where parcel-wise saliences of $BSR > 2$ are highlighted. Red indicates a positive association whereas blue indicates a negative association; i. Loadings of the individual traits (*SD: Sleep duration, D: Depression, B: BMI, I: Intelligence, SQ: Sleep quality*); ii. Relative distribution of positive (P) and negative (N) $-2 > BSR > 2$ scores per functional networks⁵³, and average BRS in functional networks⁵³ (V=visual, SM=sensorimotor, Da=dorsal-attention, Va=ventral attention, L=limbic, FP=frontopolar, DMN=default mode network), iii. Replication of brain – behavior saliency association in the HCP sample; and B. Relation between brain and behavioral saliences in HCP sample of the second brain saliency. i. Loadings of the individual traits; ii. Relation to functional networks⁵³ and iii). Relation between brain and behavioral saliences of second factor in the HCP sample.

(17) Table S2 and S3, can the significance threshold be repeated in table legend please?

We thank the reviewer for the comment and have added the significance threshold in the legend: (FDR q <0.05).

There are quite a few typos, (For example, “sleep deprivation influences frontal executive functions in both health individuals and patients. In addition, sleep deprivation disrupts leads to lower metabolism in the frontal cortex”) and the manuscript could do with a final proof read for such errors.

Many thanks for noticing these lapsus calami. The sentence has now been revised in page 20. Also, we have carefully checked the whole manuscript for such grammatical typos.

Reviewer #2

The paper examines the interrelations between sleep indices, BMI, intelligence, depression, behavioral markers and cortical thickness measures at the phenotypic and genetic levels using the HCP and NKI samples.

We are very grateful for these constructive and useful suggestions, which we incorporated in the revised version.

(1) Many testings were performed and it's not clear whether multiple testing corrections were performed appropriately. P-values should be provided in the result tables in the main text and supplementary information. It seems that the robust findings were those discovered in the literature already. The trend findings seem novel but it's not clear their robustness. A number of places need clarifications.

We thank the reviewer for this important remark. We agree that the link between sleep, mental, and physical health have been demonstrated in previous studies using small samples separately and have discussed previous observations in the introduction. However, our study is the first to address these associations comprehensively in two large-scale (both samples $n > 700$) samples of individuals integrating behavioral and neuroimaging assessments. Moreover, taking advantage of the twin structure in the HCP dataset we studied, for the first time, whether the identified phenotypic relationships between sleep and local brain structure were driven by shared additive genetic factors. We have now added this explicitly in the introduction in page 4:

“This raises the question whether the interrelation of sleep, mental, and physical health could be linked to shared neurobiological mechanisms and whether these relationships are driven by shared genetic mechanisms”

As the study is data-driven in nature and contained various analysis steps in both behavioral and cortical thickness analysis, we used an independent large-scale sample²⁰ to further validate consistency, generalizability, and replicability of (phenotypic) observations observed in the HCP-sample. Thus, as such, we believe that our study and the findings can contribute to the already existing literature. For each step in our analysis we performed multiple-comparison analysis and highlight it in each figure/table. In addition, we have added exact p-values to Table 1, all figures as well as all supplementary tables. For further clarification we have added this in the methods in page 32:

“Each analysis step is corrected for multiple comparisons. Here we consider number of tests for each sleep measure (2), as well as number of parcels (200) in case of whole brain analysis.”

(2) How was the common environmental component controlled? The genetic correlation between sleep duration and BMI may be confounded by the common/family environmental component. The authors described on page 31 “SOLAR uses maximum likelihood variance-decomposition methods to determine the relative importance of familial and environmental influences on a phenotype by modeling the covariance among family members as a function of genetic proximity.” Was the familial component included in the model?

We thank the reviewer for this question. The HCP is an extended family sample and we have used the empirical pedigree derived from their genetic data (see further²¹) to determine the kinship structure in the sample. Using the genetic data (G) and environmental data (E) genetic correlation and heritability is

computed. In the past we evaluated G+C+E (C=common/family environment component) model by including common household matrix but observed that in extended family samples, G+E is more parsimonious and leads to more reproducible results²². We have now included this in the discussion:

“In previous work²², we have shown that a model based on genetic and environmental factors only was more parsimonious as a model including common household effects in extended family samples such as the HCP sample. Nevertheless, as individual variance in sleep duration and quality was only in part explained by genetic factors, future longitudinal models might help uncover relevant familial and non-familial environmental effects relating sleep to mental and physical health.”

(3) Whether singletons were included in the genetic correlation or heritability analyses using SOLAR?

Yes, the empirical kinship matrix was created from the entire HCP sample including singletons.

(4) How was the kinship matrix estimated if singletons were contained and didn't have the pedigree information?

Many thanks for pointing out this issue. The empirical kinship matrix was created using KING method²¹.

In short (based on the report of Kochunov et al, 2019²¹), the Kinship-based INference for Genome wide association study (KING) method was developed to approximate self-reported coefficients of relationship CR values. It is frequently used to verify self-reported relationships in family samples²³. In our study, KING was used to verify zygosity for same-gendered twins. The genotyping data for 1,141 subjects were released by HCP and available through the dbGAP repository (https://www.ncbi.nlm.nih.gov/projects/gap/cgi-bin/study.cgi?study_id=phs001364.v1.p1). Briefly, all subjects were genotyped using the Illumina Multi-Ethnic Global Array²⁴ SNP-array that included chip-specific content from PsychChip and ImmunoChip and provides extended coverage of European, East Asian, and South Asian populations. We used 1,580,642 genotyped SNPs that satisfied the quality control conditions: excluding SNPs with MAF <1%, genotype call rate <95%, and Hardy–Weinberg equilibrium <1 × 10⁻⁶. The genotype data were converted to PLINK file format. The robust KING method was developed for fast $r_{i,j}$ calculations in familial samples. The method is described in details in the original publication²³. Briefly, the coefficients of relatedness are calculated using Equation (1).

$$r_{i,j} = 1 - \frac{N_{Aa}^i + N_{Aa}^j}{2N_{Aa}^i} + \frac{N_{Aa,Aa} - 2N_{AA,aa}}{N_{Aa}^i} \quad (1)$$

where, $N_{Aa}^{i,j}$ is the total number of heterozygotes for the i th and j th individuals, and $N_{Aa,Aa}$ and $N_{AA,aa}$ are the total number of SNPs at which both individuals of the pair are hetero- and homozygous. The KING method is computationally efficient because the N coefficients are computed using binary logic operations (AND and OR).

We have now added further information on the construction of the kinship matrix in the methods.

“Coefficient of relationship (genetic proximity) between individuals in the HCP sample was computed using the KING method in the openly available genotyped data of HCP. The method is described in detail in the original publication²³ and evaluated in the context of the current sample as described previously²¹.”

(5) Did the authors use their genotype data?

We thank the reviewer for this question. Indeed, the genotype data was used to create the empirical kinship matrix. Please see our answer to question 4 for more details on the kinship matrix creation.

(6) Please provide the sample size in the Tables and p values in addition to the estimates and standard errors.

According to this suggestion, we have added sample size and p-values in the tables.

(7) Table S1 listed correlations between sleep indices and many behavior markers. P values are not provided and it's not clear how the Bonferroni correction was conducted.

We apologize for the unclarity of this analysis. In the supplementary tables (Table S2 and S3), we performed exploratory analysis to identify markers that showed a strong relationship with sleep duration/quality. Here, we corrected for the total number of tested behaviors, per sleep marker.

(8) In Figure 2, panel (i), the abbreviations are not provided. It is difficult to evaluate the results.

Sorry for being unclear. Based on this suggestion, we have updated the figure and figure legend for clarity, please also see R1Q16 for the updated. Figure and legend.

(9) The significant negative correlation between sleep duration and sleep quality is not clearly described. The authors described higher scores of sleep quality meant poorer sleep quality. Thus, this negative correlation demonstrates shorter sleep related to poorer sleep quality. I would suggest providing a scatterplot next to the histogram distributions in Figure 1(A). Further clarification would help readability.

Apologies for the unclarity. We have added a scatter plot of the negative relationship between sleep duration and global sleep quality for HCP as well as eNKI in Figure 1A and D respectively.

Figure 1. Patterns of phenotypic correlation between sleep duration and cortical thickness in HCP and eNKI samples. A) distribution of variables in the unrelated HCP subsample; B+C) phenotypic correlation of sleep duration/global sleep quality and cortical thickness; D) distribution of variables in the eNKI sample, as well as the correlation between sleep duration and global sleep quality score; E+F) phenotypic correlation of sleep duration/global sleep quality and cortical thickness. Red indicates a positive relationship, whereas blue indicates a negative phenotypical relationship between sleep and brain structure. Whole-brain findings were corrected for multiple comparisons using FDR correction ($q < 0.05$, black outline). Significant associations between sleep indices and brain structure have black outline, whereas trends ($p < 0.01$) were visualized at 60% transparency.

(10) The interpretation of environmental correlation needs to be cautious, because the

environment component includes the unique environmental component and measurement errors.

We fully agree with the reviewer and therefor have now refrained from interpreting environmental correlation of the current study in our discussion. We have now explicitly mentioned this in the discussion in page 26:

“Notably, we refrain from interpreting environmental correlations, as the environment component includes the unique environmental component and measurement errors.”

References:

- 1 Medic, G., Wille, M. & Hemels, M. E. Short- and long-term health consequences of sleep disruption. *Nat Sci Sleep* **9**, 151-161, doi:10.2147/NSS.S134864 (2017).
- 2 Kervezee, L., Kosmadopoulos, A. & Boivin, D. B. Metabolic and cardiovascular consequences of shift work: The role of circadian disruption and sleep disturbances. *Eur J Neurosci*, doi:10.1111/ejn.14216 (2018).
- 3 Steptoe, A., Peacey, V. & Wardle, J. Sleep Duration and Health in Young Adults. *Archives of Internal Medicine* **166**, 1689-1692, doi:10.1001/archinte.166.16.1689 (2006).
- 4 Dashti, H. S. *et al.* Genome-wide association study identifies genetic loci for self-reported habitual sleep duration supported by accelerometer-derived estimates. *Nature Communications* **10**, 1100, doi:10.1038/s41467-019-08917-4 (2019).
- 5 Doherty, A. *et al.* GWAS identifies 14 loci for device-measured physical activity and sleep duration. *Nature Communications* **9**, 5257, doi:10.1038/s41467-018-07743-4 (2018).
- 6 Cheng, W., Rolls, E. T., Ruan, H. & Feng, J. Functional Connectivities in the Brain That Mediate the Association Between Depressive Problems and Sleep Quality. *JAMA Psychiatry* **75**, 1052-1061, doi:10.1001/jamapsychiatry.2018.1941 (2018).
- 7 Takeuchi, H. *et al.* Shorter sleep duration and better sleep quality are associated with greater tissue density in the brain. *Sci Rep* **8**, 5833, doi:10.1038/s41598-018-24226-0 (2018).
- 8 Wild, C. J., Nichols, E. S., Battista, M. E., Stojanoski, B. & Owen, A. M. Dissociable effects of self-reported daily sleep duration on high-level cognitive abilities. *Sleep* **41**, doi:10.1093/sleep/zsy182 (2018).
- 9 Smith, S. M. *et al.* A positive-negative mode of population covariation links brain connectivity, demographics and behavior. *Nat Neurosci* **18**, 1565-1567, doi:10.1038/nn.4125 (2015).
- 10 Montag, S. E. *et al.* Association of sleep characteristics with cardiovascular and metabolic risk factors in a population sample: the Chicago Area Sleep Study. *Sleep Health* **3**, 107-112, doi:10.1016/j.sleh.2017.01.003 (2017).
- 11 Lee, J. A. & Park, H. S. Relation between sleep duration, overweight, and metabolic syndrome in Korean adolescents. *Nutr Metab Cardiovasc Dis* **24**, 65-71, doi:10.1016/j.numecd.2013.06.004 (2014).
- 12 Shim, J. & Kang, S. W. Behavioral Factors Related to Sleep Quality and Duration in Adults. *J Lifestyle Med* **7**, 18-26, doi:10.15280/jlm.2017.7.1.18 (2017).
- 13 Smagula, S. F., Stone, K. L., Fabio, A. & Cauley, J. A. Risk factors for sleep disturbances in older adults: Evidence from prospective studies. *Sleep Med Rev* **25**, 21-30, doi:10.1016/j.smr.2015.01.003 (2016).

- 14 Samu, D. *et al.* Preserved cognitive functions with age are determined by domain-dependent shifts in network responsivity. *Nature Communications* **8**, 14743, doi:10.1038/ncomms14743 (2017).
- 15 Murman, D. L. The Impact of Age on Cognition. *Semin Hear* **36**, 111-121, doi:10.1055/s-0035-1555115 (2015).
- 16 Weintraub, S. *et al.* Cognition assessment using the NIH Toolbox. *Neurology* **80**, S54-64, doi:10.1212/WNL.0b013e3182872ded (2013).
- 17 Tadayon, E., Pascual-Leone, A. & Santarnecchi, E. Differential Contribution of Cortical Thickness, Surface Area, and Gyrification to Fluid and Crystallized Intelligence. *Cereb Cortex*, doi:10.1093/cercor/bhz082 (2019).
- 18 Christoforou, A. *et al.* GWAS-based pathway analysis differentiates between fluid and crystallized intelligence. *Genes Brain Behav* **13**, 663-674, doi:10.1111/gbb.12152 (2014).
- 19 Lim, J. & Dinges, D. F. A meta-analysis of the impact of short-term sleep deprivation on cognitive variables. *Psychol Bull* **136**, 375-389, doi:10.1037/a0018883 (2010).
- 20 Glasser, M. F. *et al.* The minimal preprocessing pipelines for the Human Connectome Project. *Neuroimage* **80**, 105-124, doi:10.1016/j.neuroimage.2013.04.127 (2013).
- 21 Kochunov, P. *et al.* Genomic kinship construction to enhance genetic analyses in the human connectome project data. *Hum Brain Mapp* **40**, 1677-1688, doi:10.1002/hbm.24479 (2019).
- 22 Kochunov, P. *et al.* Homogenizing Estimates of Heritability Among SOLAR-Eclipse, OpenMx, APACE, and FPHI Software Packages in Neuroimaging Data. *Front Neuroinform* **13**, 16, doi:10.3389/fninf.2019.00016 (2019).
- 23 Manichaikul, A. *et al.* Robust relationship inference in genome-wide association studies. *Bioinformatics* **26**, 2867-2873, doi:10.1093/bioinformatics/btq559 (2010).
- 24 Wegbreit, E. *et al.* Developmental Meta-analyses of the Functional Neural Correlates of Bipolar Disorder. *JAMA Psychiatry* **71**, 926-935, doi:10.1001/jamapsychiatry.2014.660 (2014).

Reviewers' comments:

Reviewer #1 (Remarks to the Author):

The authors have addressed all my comments

Reviewer #2 (Remarks to the Author):

The authors have addressed some of my concerns. My concern is still whether false positives are properly controlled (i.e., if the results are robust). The authors performed many tests. Multiple comparison corrections are only adjusted at each individual step, and there are many steps and analyses across the whole study. Although the authors use an independent sample as a way for validation, the results of brain maps are not consistent between the two samples.

In my opinion, the paper can be more concise and the clarity can be improved. For example, in Table 1, it seems to me many of these values are phenotypic correlations. It was not clear from the figure legend or the text. The authors use “* indicates $p < 0.05$, ** indicates FDR $q < 0.05$.”, but this rule does not seem to be consistent throughout the table. The authors state that “Depression, IQ, and BMI were all significantly heritable (Table 1) and we observed a negative genetic correlation between BMI and IQ ($\rho_g = -0.27$ (0.06)).” This genetic correlation ($\rho_g = -0.27$) cannot be found from the table.

There are typos in different places.

Reviewers' comments:

Reviewer #2

1. The authors have addressed some of my concerns. My concern is still whether false positives are properly controlled (i.e., if the results are robust). The authors performed many tests. Multiple comparison corrections are only adjusted at each individual step, and there are many steps and analyses across the whole study. Although the authors use an independent sample as a way for validation, the results of brain maps are not consistent between the two samples.

We thank the reviewer for restating these important remarks. The current study takes an exploratory approach to assess the interrelationship between sleep, mental, and physical health, and local brain structure. We have now included this explicitly in the introduction, p. 4:

“...to explore whether the interrelation of sleep, mental, and physical health can be linked to shared neurobiological mechanisms.”

Importantly, we consistently corrected for multiple comparisons within each analysis step, as, considering the inter-correlation across the tested measures (e.g. measures of sleep quality and duration, BMI, IQ, and depression), correcting for multiple comparisons across all the analyses steps would have been too conservative and could result in false negatives. For this reason, we assessed associations between sleep and local brain structure, while controlling for the other related factors. Second, to control for false positives and sample specific effects, we repeated our main analyses in an independent sample with a broad age range, the enhanced NKI sample. Moreover, we performed various post-hoc stability assessments, testing associations in different age ranges and short vs. long sleep.

In order to be clear about the discrepancy between samples regarding the relation between sleep and local brain structure, we have now highlighted the inconsistent univariate relationship between local cortical thickness and sleep also in the paper abstract:

“Sleep was associated with thickness in frontal, temporal, and occipital cortices, however local associations were inconsistent across samples.”

Considering the inter-relationship between the different measures tested in the current study, we evaluated associations between cortical thickness and all considered behavioral scores, using a multivariate approach (partial least squares). It has been suggested that multiple comparison corrections in mass univariate analysis may result in missing effects and thus inconsistencies in the results¹. Indeed, the results of this analysis indicate that multivariate patterns in distributed regions together contribute to latent behavioral factors underlying sleep, mental and physical health. These findings were observed in both the eNKI and HCP sample, suggesting that though the strength of the univariate association between sleep and thickness varied between samples, comparable and consistent multivariate relationships between local cortical thickness and sleep and health factors exist in two independent samples.

In sum, in two independent samples we observed diverging univariate but consistent multivariate associations between sleep and local cortical thickness. We have discussed possibly reasons for divergence between samples in the discussion and underlined that local, univariate, associations between sleep and cortical thickness should be interpreted with caution (p. 22).

“Possible causes for divergence could be sample characteristics, as well as confounding effects. However, even when controlling for age, intelligence, BMI, and depression, findings remained dissimilar between samples. Only when evaluating spatial patterns of relationships between sleep duration and cortical

thickness, we observed cross-sample consistency, suggesting that the degree of impact of sleep duration on local brain structure varied across samples, but that the direction of the relation between sleep and cortical thickness was comparable across the cortex. It is of note that though we observed diverging phenotypic relationships across samples, three out of four local relationships between sleep and cortical thickness were observed driven by additive genetic factors, suggestive of a system-level impact of sleep on brain structure, with modest but robust underlying local genetic associations. Nevertheless, local, univariate, associations between sleep and cortical thickness should be interpreted with caution, as they were not consistent across samples.”

Moreover, to further support research on this question, we have made our analysis scripts openly available at GitHub.

2. In my opinion, the paper can be more concise and the clarity can be improved. For example, in Table 1, it seems to me many of these values are phenotypic correlations. It was not clear from the figure legend or the text. The authors use “* indicates $p < 0.05$, ** indicates FDR $q < 0.05$.”, but this rule does not seem to be consistent throughout the table.

We thank the reviewer for noting this unclarity. We have now edited the table 1 and table legend to be more concise.

Sleep duration ($h^2 = 0.24 \pm 0.06$)			
Sample	Depression ($h^2 = 0.24 \pm 0.06$)	BMI ($h^2 = 0.68 \pm 0.04$)	IQ ($h^2 = 0.66 \pm 0.04$)
HCP (unrelated sample)	(n=419) -0.09 [-0.19 - 0.00], $p = 0.06$	(n=424) -0.11 [-0.21 - 0.02], $p < 0.025$ *	(n=418) 0.11 [0.01 0.19], $p < 0.05$ *
eNKI	(n=782) -0.16 [-0.24 - 0.09], $p < 0.001$ **	(n=757) -0.17 [-0.24 - 0.09], $p < 0.001$ **	(n=783) 0.11 [0.04 0.18], $p < 0.005$ *
HCP (total sample)	(n=1105) -0.07 [-0.13 - 0.02], $p < 0.025$ *	(n=1112) -0.14 [-0.19 - 0.08], $p < 0.0001$ **	(n=1096) 0.09 [0.03 0.15], $p < 0.005$ *
Genetic correlation (HCP)	0.17(0.20), $p > 0.1$	-0.33 (0.11), $p < 0.005$ *	0.42 (0.11), $p < 0.0001$ **
Environmental correlation (HCP)	-0.16(0.06), $p < 0.01$ *	0.01 (0.07), $p > 0.1$	0.19 (0.06), $p < 0.003$ **
Global sleep quality ($h^2 = 0.12 \pm 0.06$)			
Sample	Depression	BMI	IQ
HCP (unrelated sample)	(n=419) 0.37 [0.29 0.45], $p < 0.0001$ **	(n=424) 0.14 [0.04 0.23], $p < 0.005$ *	(n=418) -0.07 [-0.16 0.03], $p > 0.1$
eNKI	(n=782) 0.31 [0.25 0.38], $p < 0.0001$ **	(n=757) 0.09 [0.02 0.17], $p < 0.01$ *	(n=419) -0.09 [-0.16 - 0.02], $p < 0.01$ *
HCP (total sample)	(n=1112) 0.35 [0.30 0.40], $p < 0.0001$ **	(n=1112) 0.10 [0.04 0.16], $p < 0.001$ **	(n=1096) -0.10 [-0.16 - 0.04], $p < 0.002$ *
Genetic correlation (HCP)	0.32(0.26), $p > 0.1$	0.41 (0.16), $p < 0.025$ *	-0.59 (0.20), $p < 0.0001$ **
Environmental correlation (HCP)	0.38(0.05), $p < 0.0001$ **	0.03 (0.07), $p > 0.1$	0.17 (0.06), $p < 0.007$ **

Table 1. Phenotypic and genetic correlations between sleep and depression, BMI, and IQ. We performed phenotypic (HCP unrelated sample, eNKI sample, HCP total sample) and genetic correlation (HCP total sample) analysis of the association between sleep duration and global sleep quality on the one hand, and depression, BMI, and IQ on the other, including 95% confidence intervals. ** indicates FDR $q < 0.05$ and * indicates association at trend-level $p < 0.05$. Sample sizes are reported for each analysis.

3. The authors state that “Depression, IQ, and BMI were all significantly heritable (Table 1) and we observed a negative genetic correlation between BMI and IQ ($\rho_g = -0.27$ (0.06)).” This genetic correlation ($\rho_g = -0.27$) cannot be found from the table.

We apologize for the unclarity, the results of phenotypic and genetic correlation between non-sleep variables were reported in the “Supplementary Table 4”, we have now added a reference to this table in the manuscript.

4. There are typos in different places.

We thank the reviewer for this comment and two independent readers have carefully revised the manuscript before resubmission.

- 1 Kharabian Masouleh, S., Eickhoff, S. B., Hoffstaedter, F., Genon, S. & Alzheimer's Disease Neuroimaging, I. Empirical examination of the replicability of associations between brain structure and psychological variables. *elife* (2019).

REVIEWERS' COMMENTS:

Reviewer #2 (Remarks to the Author):

I have no more comments